# EphrinB1 modulates glutamatergic inputs into POMC-expressing progenitors and controls glucose homeostasis

**Manon Gervais, Gwenaël Labouèbe[☉], Alexandre Picard[iD][☉], Bernard Thorens[iD], Sophie Croizier[iD]\***

Center for Integrative Genomics, University of Lausanne, Lausanne, Switzerland

☉ These authors contributed equally to this work.
\* sophie.croizier@unil.ch

**Data Availability Statement:** The source data underlying Figs 1C, 1F–1K; 2A and 2C; 3C and 3D–3F, 4B and 4C; 5C and 5D; 6A–6F, 6H and 6J; 7A, 7B, 7D–7M; S1E–S1G Fig; S3A–S3D Fig; S4B–S4K

## Abstract

Proopiomelanocortin (POMC) neurons are major regulators of energy balance and glucose homeostasis. In addition to being regulated by hormones and nutrients, POMC neurons are controlled by glutamatergic input originating from multiple brain regions. However, the factors involved in the formation of glutamatergic inputs and how they contribute to bodily functions remain largely unknown. Here, we show that during the development of glutamatergic inputs, POMC neurons exhibit enriched expression of the *Efnb1* (EphrinB1) and *Efnb2* (EphrinB2) genes, which are known to control excitatory synapse formation. In vivo loss of *Efnb1* in POMC-expressing progenitors decreases the amount of glutamatergic inputs, associated with a reduced number of α-amino-3-hydroxy-5-methyl-4-isoxazolepropionic acid (AMPA) and N-methyl-D-aspartate (NMDA) receptor subunits and excitability of these cells. We found that mice lacking *Efnb1* in POMC-expressing progenitors display impaired glucose tolerance due to blunted vagus nerve activity and decreased insulin secretion. However, despite reduced excitatory inputs, mice lacking *Efnb2* in POMC-expressing progenitors showed no deregulation of insulin secretion and only mild alterations in feeding behavior and gluconeogenesis. Collectively, our data demonstrate the role of ephrins in controlling excitatory input amount into POMC-expressing progenitors and show an isotype-specific role of ephrins on the regulation of glucose homeostasis and feeding.

## Introduction

Obesity and associated diseases, such as type 2 diabetes, are major public health concerns, and their worldwide prevalence is increasing at an alarming rate. Energy and glucose homeostasis are centrally controlled by complex neuronal networks that involve 2 main antagonistic neuronal populations in the arcuate nucleus of the hypothalamus (ARH): the anorexigenic proopiomelanocortin (POMC) neurons and the orexigenic Agouti-related peptide (AgRP)/neuropeptide Y (NPY) coexpressing neurons [1–4]. In addition to the key role they perform in controlling feeding, POMC and AgRP/NPY neurons are involved in the control of glucose homeostasis [5,6]. Indeed, insulin action on AgRP neurons suppresses hepatic glucose

Fig, S5A–S5J Fig are provided in separated Excel spread sheets in S1 Data. RNA-seq data are available in a public repository (GEO). Accession # GSE144887.

**Funding:** This work was supported by the Swiss National Science Foundation grant (PZ00P3_167934/1) and Novartis grant (19B145) (to SC) and by an European Research Council Advanced Grant (INTEGRATE, No. 694798) and by a grant from the Swiss National Science Foundation (310030-182496) (to BT). The funders had no role in study design, data collection and analysis, decision to publish, or preparation of the manuscript.

**Competing interests:** The authors have declared that no competing interests exist.

**Abbreviations:** AAV, adeno-associated virus; ACTH, adrenocorticotropic hormone; AgR, Agouti-related peptide; AMPA, α-amino-3-hydroxy-5-methyl-4-isoxazolepropionic acid; ANOVA, analysis of variance; ARH, arcuate nucleus of the hypothalamus; CPM, count per million; FACS, fluorescence-activated cell sorting; FDR, false discovery rate; GFP+, GFP-positive; GTT, glucose tolerance test; hrGFP, humanized Renilla GFP; ITT, insulin tolerance test; IHC, immunohistochemistry; IP, intraperitoneal; NMDA, N-methyl-D-aspartate; NMR, nuclear magnetic resonance; NPY, neuropeptide Y; PCA, principal component analysis; POMC, proopiomelanocortin; PTT, pyruvate tolerance test; PVH, paraventricular nucleus of the hypothalamus; RNA-seq, RNA sequencing; RT-PCR, quantitative reverse transcription PCR; sEPSC, spontaneous excitatory postsynaptic currents; SEM, standard error of the mean; WT, wild-type.

production [7,8], and chronic activation and inhibition of POMC neurons represses and stimulates gluconeogenesis, respectively [9]. Moreover, alterations in POMC signaling, circuits, or neuronal survival are associated with the disruption of glucose homeostasis [10–12].

POMC and AgRP/NPY neurons are major integrators of peripheral hormones (insulin, leptin, and ghrelin) and nutrients (glucose) to control energy and glucose homeostasis through neuroendocrine and autonomic responses [13–17]. Besides, POMC and AgRP/NPY neurons also receive abundant central information through GABAergic (inhibitory) and glutamatergic (excitatory) inputs [18,19]. Notably, POMC neurons primarily receive glutamatergic inputs, whereas AgRP/NPY neurons receive primarily GABAergic inputs [20]. However, the mechanisms underlying the development of POMC neuronal circuits and particularly the formation of glutamatergic inputs and how they contribute to glucose homeostasis and energy balance remain elusive.

During the development of the nervous system, cues that guide development orchestrate neuronal wiring to allow growing axons to reach their targets and establish synaptic contacts to build functional neuronal networks. EphrinB molecules form a family of cell-contacting proteins that specifically interact with EphA and EphB receptors [21] to stabilize glutamatergic synapses, to recruit α-amino-3-hydroxy-5-methyl-4-isoxazolepropionic acid (AMPA) and N-methyl-D-aspartate (NMDA) receptors, and to control the number of glutamatergic synapses [22–26]. In the rat ARH, synapse formation begins postnatally and gradually increases until adulthood [27]. However, the molecules underlying this developmental process are still unknown.

Here, we employed a transcriptomic approach to reveal that the *Efnb1* and *Efnb2* gene products (EphrinB1 and EphrinB2) are enriched in POMC neurons when the development of glutamatergic inputs occurs. In mice, the lack of *Efnb1* or *Efnb2* in POMC-expressing progenitors (POMC^prog) decreases the amount of glutamatergic inputs, affects the expression of AMPAR and NMDAR subunits, and decreases the amplitude and the frequency of the spontaneous excitatory postsynaptic currents (sEPSC) of POMC-expressing cells but has isotype-specific metabolic effects impacting glucose tolerance, vagus nerve activation, and insulin secretion (*Efnb1* deletion) or feeding and gluconeogenesis (*Efnb2* deletion). Our data show that POMC^prog belong to a complex neuronal network and can integrate central and peripheral information to control glucose homeostasis.

## Results

### Onset of glutamatergic inputs into POMC-expressing progenitors

To visualize the development of excitatory presynaptic terminals in POMC^prog, we performed immunohistochemical labeling of presynaptic glutamatergic vesicular transporter (vGLUT2) in POMC^prog labeled with a tdTomato reporter (*Pomc*-Cre;tdTomato) in postnatal days P6, P14, and P22 male mice (Fig 1A). The analysis was exclusively focused on tdTomato-positive neurons found in the dorsal and lateral parts of the ARH where most of the POMC neurons are expressed. At P6, glutamatergic inputs were already observed to be in contact with POMC^prog, and the amount of inputs increased gradually until P22 (Fig 1A–1C).

Based on this developmental observation and knowing that POMC progenitors can give rise to NPY, POMC [28], and Kisspeptin neurons [29], we next performed transcript profiling of POMC^prog that do not express NPY (enriched in POMC+ neurons) and NPY+ cells at P14 to identify putative genes involved in glutamatergic synapse formation (Fig 1D and 1E). We thus performed RNA sequencing (RNA-seq) experiments mostly on POMC neurons (i.e., POMC->POMC), on NPY neurons derived from POMC progenitors (i.e., POMC->NPY), or on NPY neurons not derived from POMC progenitors (i.e., NPY->NPY). These 3 subpopulations

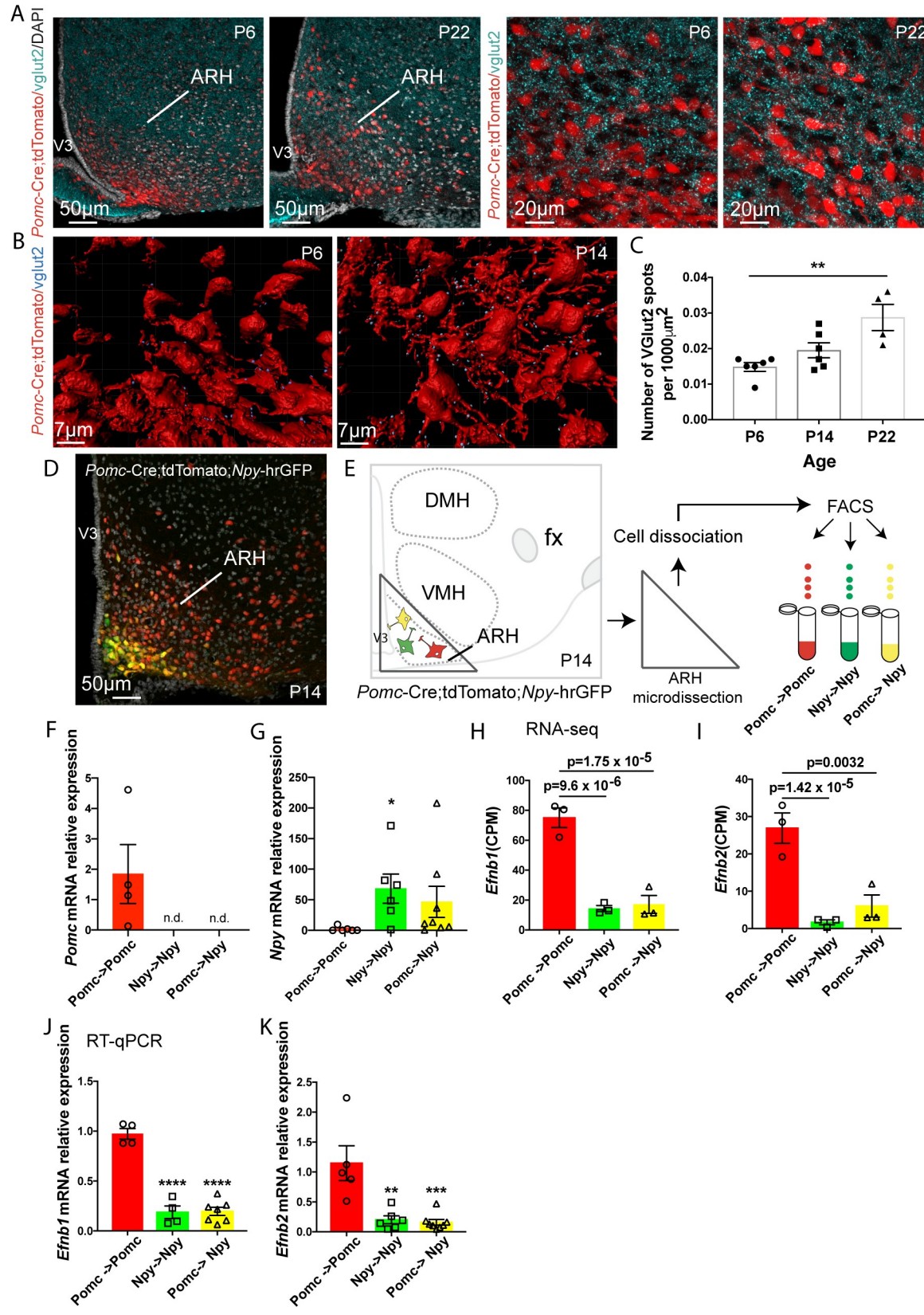

**Fig 1. Enrichment of *Efnb1* and *Efnb2* mRNA in POMC neurons during postnatal development of glutamatergic inputs.** (A) Confocal images and high magnification of vGLUT2-positive terminals (turquoise)/DAPI (white) into *Pomc*-Cre;tdTomato neurons (red) in the ARH of P6 and P22 male mice. (B) Example of 3D reconstruction (P6 and P22, IMARIS) and quantification (C) of vGLUT2-positive inputs in direct apposition with *Pomc*-Cre;tdTomato neurons of P6, P14, and P22 male pups (*n* = 2–3/group, 2 sections/animal). (D, E) Image and schematic illustrating *Pomc*-Cre;tdTomato progenitors (Pomc->Pomc, red) and their partial co-expression with *Npy*-hrGFP neurons (Pomc->Npy, yellow) in the ARH of P14 male mice. *Npy*-hrGFP neurons not deriving from Pomc progenitors (Npy->Npy) are labeled in green. These 3 sub-populations are sorted by flow cytometry based on their fluorescence (E). *Pomc* (F) and *Npy* (G) mRNA relative expression in Pomc->Pomc, Pomc->Npy and Npy->Npy populations. RNA-seq data (CPM) highlighted *Efnb1* (H) and *Efnb2* (I) genes as enriched in Pomc->Pomc neurons when compared to Npy->Npy and Pomc->Npy. *Efnb1* (J) and *Efnb2* (K) mRNA expression in Pomc->Pomc, Npy->Npy, and Pomc->Npy populations measured by RT-qPCR. Data are shown ± SEM. Statistical significance was determined using 1-way ANOVA test (C, F, G, J, K). $^*P \leq 0.05$ versus Pomc->Pomc (F), $^{**}P \leq 0.01$ versus P6 (C), versus Pomc->Pomc (K); $^{***}P \leq 0.001$ versus Pomc->Pomc (K); $^{****}P \leq 0.0001$ versus Pomc->Pomc (J). The underlying data are provided in S1 Data. ANOVA, analysis of variance; ARH, arcuate nucleus of the hypothalamus; CPM, count per million; DMH, dorsomedial nucleus of the hypothalamus; fx, fornix; hrGFP, humanized Renilla GFP; NPY, neuropeptide Y; POMC, proopiomelanocortin; RNA-seq, RNA sequencing; RT-qPCR, quantitative reverse transcription PCR; SEM, standard error of the mean; VMH, ventromedial nucleus of the hypothalamus; V3, third ventricle.

can be separated based on their fluorescence when using *Pomc*-Cre;tdTomato;*Npy*-humanized Renilla GFP (hrGFP) animals (red cells: POMC->POMC; yellow cells: POMC->NPY cells; and green cells: NPY->NPY) (Fig 1D–1G and S1A and S1B Fig). To validate the quality of the sort, we assessed by quantitative reverse transcription PCR (RT-qPCR) the relative level of expression of *Pomc* and *Npy* mRNA in POMC->POMC, POMC->NPY, and NPY->NPY populations. As expected, *Pomc* mRNA is highly expressed in POMC->POMC and undetectable in POMC->NPY and NPY->NPY neurons (Fig 1F). On the contrary, *Npy* mRNA is enriched in POMC->NPY and NPY->NPY neurons when compared with that in POMC->POMC population (Fig 1G). The principal component analysis (PCA) of the RNA-seq data reveals that POMC->NPY and NPY->NPY populations seem to be distinguishable, but because of the outlier sample P14.33 NPY->NPY, this difference is not significant (S1 Fig). To facilitate our study, we only focused our attention on results obtained from POMC->POMC and NPY->NPY neurons. This analysis revealed that 1,586 genes were up-regulated, and 1,202 genes were down-regulated in POMC->POMC neurons compared with those in NPY->NPY neurons. However, when the analysis was restricted to genes involved in axon guidance and synaptogenesis (S1 Fig and S1 Table), out of the 37 identified genes, 2 were significantly up-regulated and 4 were down-regulated in POMC neurons compared with those in NPY neurons. In particular, *Efnb1* and *Efnb2* were 5.2- and 16-fold enriched in POMC->POMC neurons when compared with those in NPY->NPY neurons, respectively (Fig 1H and 1I and S1 Fig). We further confirmed this increase in *Efnb1* and *Efnb2* mRNA expressions in POMC^prog devoid of NPY neurons using RT-qPCR and observed consistent results; there was a 5.2- and 5.8-fold increase in *Efnb1* and *Efnb2* mRNA, respectively (Fig 1J and 1K). To determine whether *Efnb1* and *Efnb2* were differentially expressed in POMC neurons in particular based on their anatomical location, we performed in situ hybridization and quantified the number of fluorescent punctate signals in *Pomc*-GFP-positive (GFP^+) neurons in distinct anteroposterior parts of the ARH of P14 animals (S1 Fig). This analysis revealed that POMC neurons homogeneously expressed *Efnb1* and *Efnb2* throughout the entire ARH (S1 Fig).

## EphrinB members are expressed during development of hypothalamic glutamatergic projections onto POMC-expressing progenitors

To study whether EphrinB1 and EphrinB2 can directly modulate the number of glutamatergic terminals on POMC neurites, we had to identify the glutamatergic areas innervating POMC neurons. Previous monosynaptic retrograde mapping showed that POMC^prog receive inputs from several sites, such as the preoptic area, the bed nucleus of the stria terminalis [19], the

lateral septum [19], and, in particular, the paraventricular nucleus of the hypothalamus (PVH) [18]. The PVH is known to contain glutamatergic neurons [17]; however, whether glutamatergic PVH neurons innervate ARH POMC neurons remain elusive. To address this question, we used a retrograde viral approach using modified rabies virus SADΔG-mcherry combined with Cre-dependent helper adeno-associated virus (AAV) (S2 Fig). An injection of these viruses into the ARH of *Pomc*-Cre male mice allows a retrograde monosynaptic spread from POMC-prog. We combined the detection of retrogradely labeled mCherry-positive cells with in situ hybridization of *vglut2* mRNA. This allowed us to visualize PVH cells that were retrogradely labeled with mCherry and were glutamatergic (S2 Fig). These observations confirmed that POMCprog receive glutamatergic inputs from the PVH.

We next examined whether ephrin receptors EphB1, EphB2, EphB3, EphB4, EphA4, and EphA5 were expressed by presynaptic neurons of the PVH when glutamatergic terminals developed into POMCprog (from P8 to P18). We found that *Ephb1*, *Ephb2* (Fig 2A), *Ephb3*, *Ephb4*, *Epha4*, and *Epha5* mRNA (S3 Fig) were expressed in microdissected PVH during this postnatal period. *Ephb1* and *Ephb2* are well-known receptors of EphrinB1 and EphrinB2 [30]. We thus assessed by in situ hybridization whether *Ephb1* and *Ephb2* mRNA were specifically expressed by glutamatergic neurons of the PVH at P14. Our results show that 97.2% and 88.2% of glutamatergic presynaptic neurons in the PVH expressed *Ephb1* and *Ephb2*, respectively (Fig 2B and 2C). These data suggest that the establishment of PVH glutamatergic inputs into POMCprog may depend on presynaptic EphB receptors and postsynaptic EphrinB (Fig 2D).

## Lack of *Efnb1* in POMC-expressing progenitors is associated with impaired glucose homeostasis

To determine whether *Efnb1* is required for the normal development of glutamatergic inputs on POMCprog in vivo, we crossed mice carrying an *Efnb1*loxP allele [31] with mice expressing

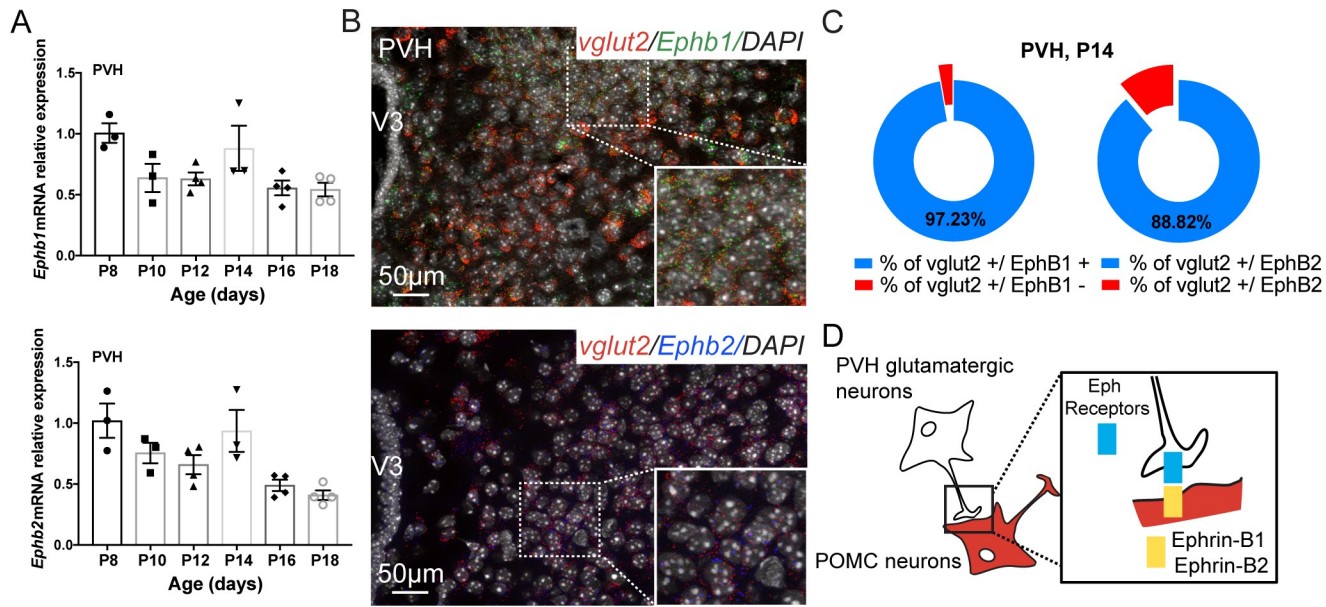

**Fig 2. EphrinB signaling.** (A) Quantification of *Ephb1* and *Ephb2* mRNA relative expressions in the PVH of P8 to P18 male mice (*n* = 3–4/group). Photomicrographs (B) and quantification (C) showing co-expression of *Ephb1* (green) and *Ephb2* (blue) mRNA with *vglut2* mRNA (red, glutamatergic marker) in the PVH of P14 male mice. DAPI counterstaining is shown in white. (D) Schematic of our working model. Data are shown ± SEM. Statistical significance was determined using 1-way ANOVA (A). The underlying data are provided in S1 Data. ANOVA, analysis of variance; POMC, proopiomelanocortin; PVH, paraventricular nucleus of the hypothalamus; SEM, standard error of the mean; V3, third ventricle.

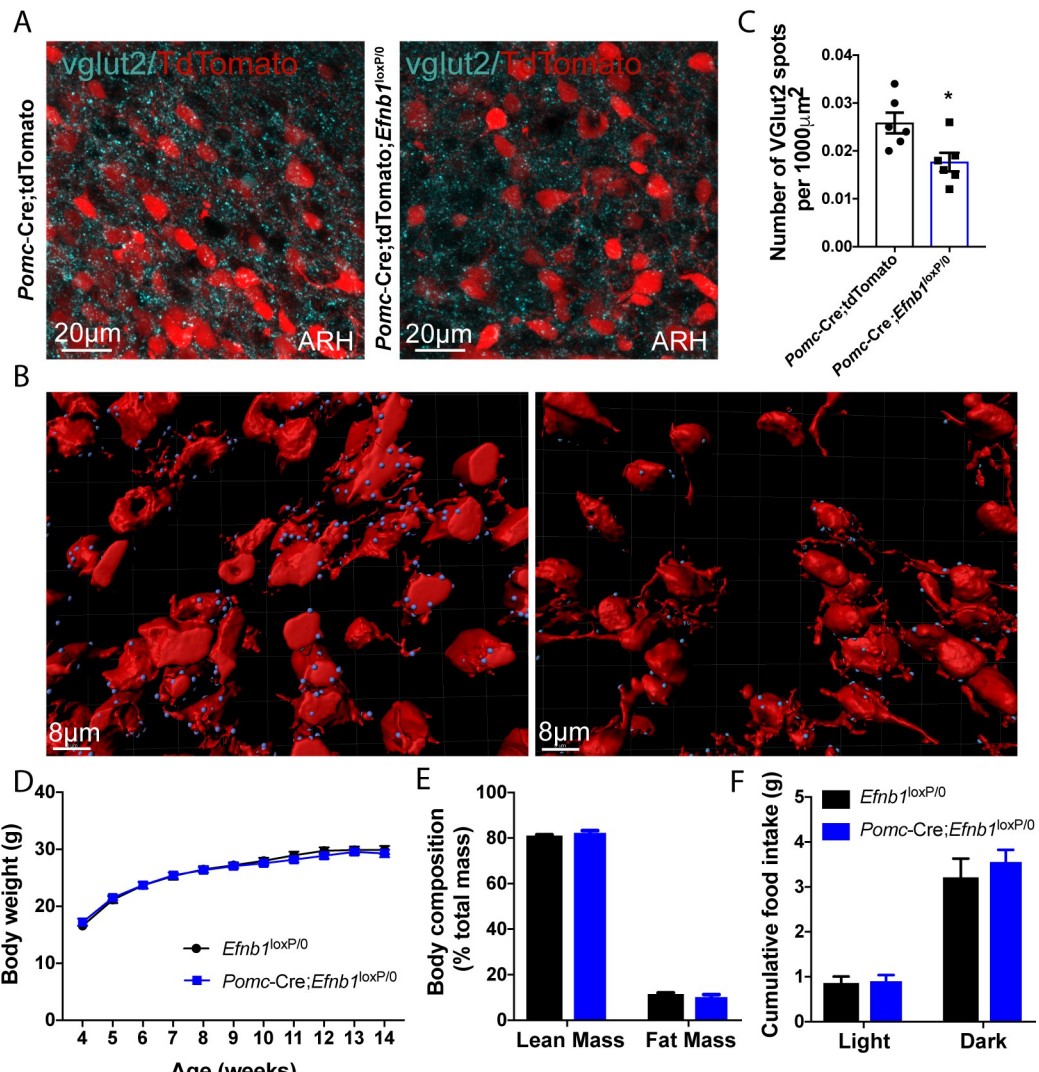

**Fig 3. Lack of *Efnb1* in POMC-expressing progenitors decreases the amount of glutamatergic inputs into these neurons.** (A) High magnification of vGLUT2-positive terminals into POMC-expressing progenitors (red) in *Pomc*-Cre; tdTomato control and *Pomc*-Cre;*Efnb1*<sup>loxP/0</sup>;tdTomato mutant male mice. (B) Example of 3D reconstruction (IMARIS) and quantification (C) of vGLUT2-positive inputs (blue) in direct apposition with *Pomc*-Cre;tdTomato neurons (red) (*n* = 3/group, 2 sections/animal). (D) Post-weaning growth curve of *Efnb1*<sup>loxP/0</sup> and *Pomc*-Cre;*Efnb1*<sup>loxP/0</sup> male mice (*n* = 14–15/group). (E) Body composition of 16–18-week-old *Efnb1*<sup>loxP/0</sup> and *Pomc*-Cre mice (*n* = 9–10/group). (F) Cumulative food intake of 13–14-week-old *Efnb1*<sup>loxP/0</sup> and *Efnb1*<sup>loxP/0</sup>;*Pomc*-Cre male mice (*n* = 9–12/group). Data are shown ± SEM. Statistical significance was determined using 2-way ANOVA (D–F) and 2-tailed Student *t* test (C). $^*P \leq 0.05$ versus *Pomc*-Cre; tdTomato (C). The underlying data are provided in S1 Data. ANOVA, analysis of variance; ARH, arcuate nucleus of the hypothalamus; POMC, proopiomelanocortin; SEM, standard error of the mean.

Cre recombinase in a *Pomc*-specific manner [32]. We observed a 31% decrease in excitatory vGLUT2<sup>+</sup> inputs into arcuate POMC-expressing progenitors found in the dorsal and lateral parts of the ARH in 16-week-old *Pomc*-Cre;*Efnb1*<sup>loxP/0</sup> mutant male mice (Fig 3A–3C). To assess whether the reduced number of glutamatergic inputs into POMC<sup>prog</sup> was associated with a decrease of glutamatergic AMPA and NMDA receptors, we labeled AMPA and NMDA subunits mRNA, *Gria1* and *Grin1*, respectively (Fig 4A). We thus quantified the number of *Gria1* and *Grin1* mRNA spots in POMC-expressing neurons in mice lacking *Efnb1* in POMC-<sup>prog</sup>. As expected, we observed a 29% and 28% respective decrease of *Gria1* and *Grin1* mRNA

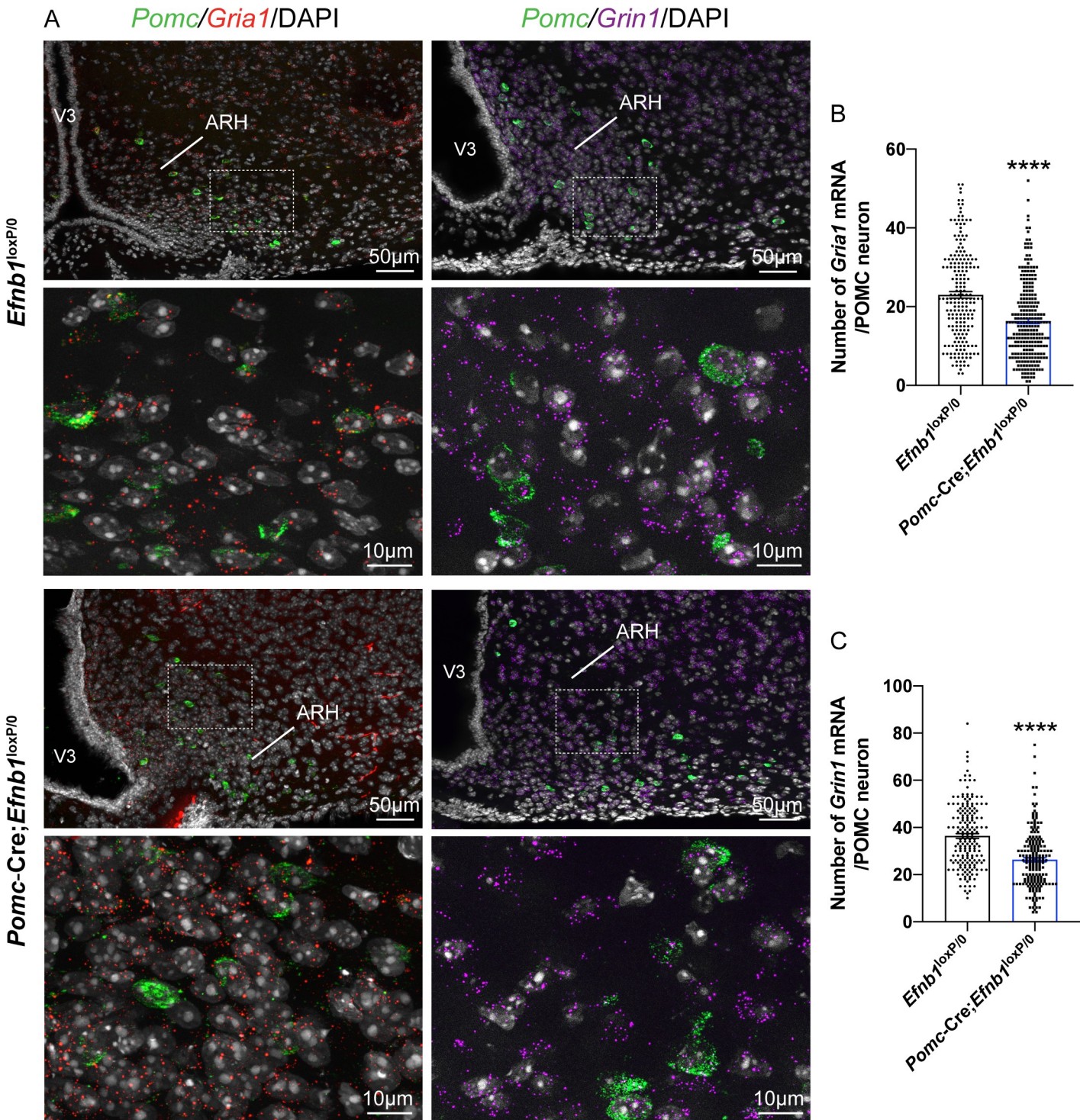

**Fig 4. *Gria1* and *Grin1* mRNA expressions in POMC neurons is impaired by the lack of *Efnb1* in POMC-expressing progenitors.** (A) Microphotographs showing *Gria1* (red) and *Grin1* (purple) mRNA spots in *Pomc*-expressing neurons (green) neurons in the ARH of 19–20-week-old male mice. DAPI counterstaining is shown in white. Quantification of the number of *Gria1* (B) and *Grin1* (C) mRNA spots in POMC neurons ($n$ = 2–3 animals/group, $n$ = 206–263 neurons/group). Data are shown ± SEM. Statistical significance was determined using 2-tailed Student $t$ test (B, C). ****$P \leq 0.0001$ versus *Efnb1*[loxP/loxP] male mice (B, C). The underlying data are provided in S1 Data. ARH, arcuate nucleus of the hypothalamus; POMC, proopiomelanocortin; SEM, standard error of the mean; V3, third ventricle.

spots in conditional mutant mice when compared with control mice (Fig 4B and 4C). To determine whether the decrease of AMPAR and NMDAR subunits is associated with an alteration of the excitability of POMC[prog], we recorded AMPAR-mediated sEPSC in tdTomato-positive cells found in dorsal and lateral parts of the ARH where most of POMC neurons are observed using electrophysiology approach (Fig 5A). The lack of *Efnb1* in POMC[prog] causes a decrease of the amplitude (Fig 5B and 5C) and the frequency (Fig 5B and 5D) of sEPSC, respectively.

Next, we examined whether the lack of *Efnb1* in POMC[prog] caused disturbances in body weight and food intake. The postnatal growth curves (body weights) and body composition of *Pomc*-Cre;*Efnb1*[loxP/0] mice were undistinguishable from *Efnb1*[loxP/0] control male mice (Fig 3D and 3E). Consistent with these data, daily food intake was similar between the groups (Fig 3F). In addition to their fundamental role in energy balance, POMC neurons have been shown to be involved in the regulation of peripheral glucose homeostasis [5,6]. Accordingly, we also investigated the effect of genetically deleting *Efnb1* in POMC[prog] on peripheral glucose homeostasis and insulin sensitivity. The basal glycemia and insulinemia were unchanged in *Pomc*-Cre;*Efnb1*[loxP/0] mice and *Efnb1*[loxP/0] mice (Fig 6A and 6B). *Pomc*-Cre;*Efnb1*[loxP/0] mice displayed significantly elevated glycemia 15 to 45 minutes after a glucose challenge (Fig 6C and

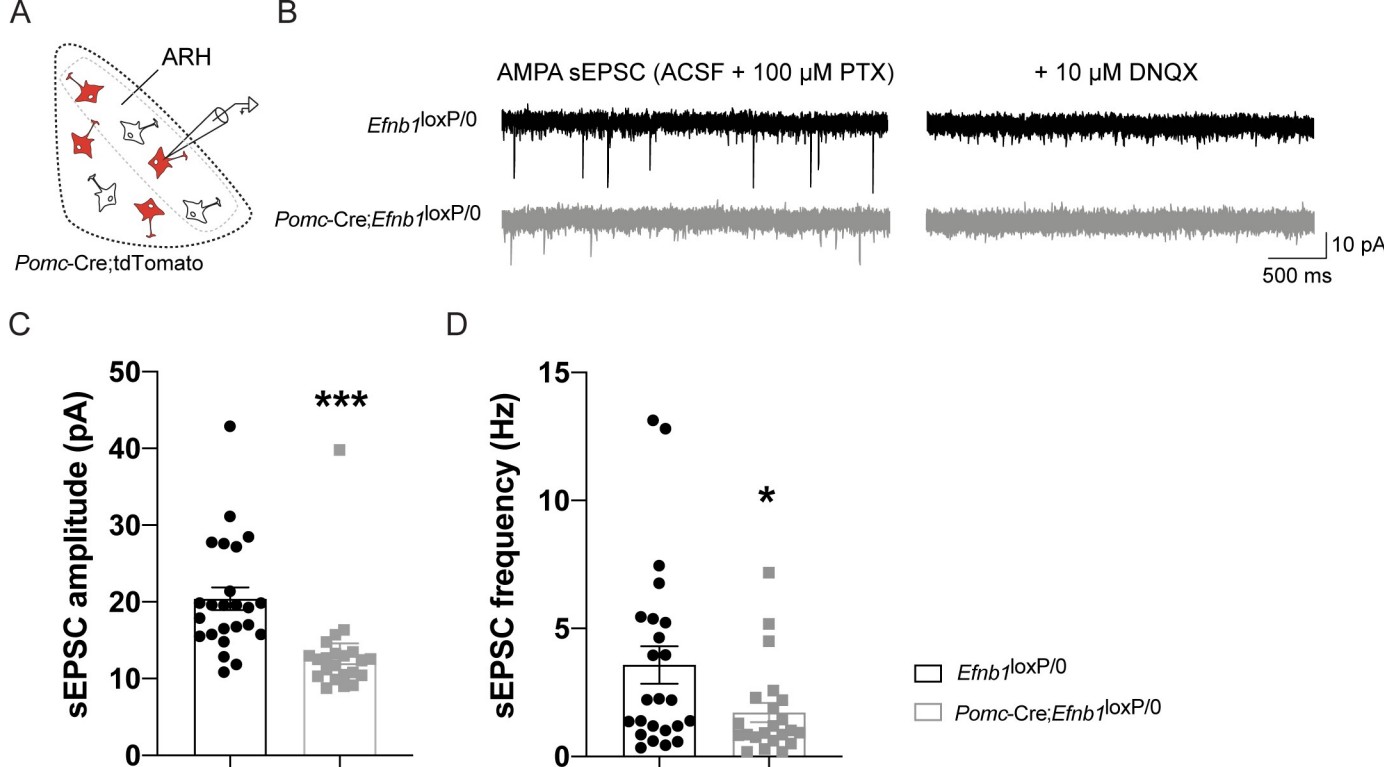

**Fig 5. AMPAR-mediated sEPSC of POMC-expressing progenitors are affected by the lack of *Efnb1*.** (A) Schematic showing the experimental approach used in panels B–D. POMC-expressing progenitor-positive neurons used for the recording were identified via tdTomato expression and found in dorsal and lateral parts of the arcuate nucleus (ARH, grey dashed line). (B) Whole-cell patch clamp monitoring of sEPSC in control *Efnb1*[loxP/0] (black traces) or *Efnb1*[loxP/0];*Pomc*-Cre (grey traces) mice. tdTomato-positive neurons were clamped at −70 mV, and the GABA$_A$ receptor antagonist picrotoxin was bath applied to isolate AMPAR sEPSC. Confirmation of AMPAR sEPSC isolation was accomplished by adding the AMPAR antagonist DNQX at the end of the experiment (right traces). Quantitative analysis of AMPA sEPSC revealed that *Efnb1* deletion led to a significant reduction of postsynaptic currents amplitude (C) and frequency (D). This suggests an alteration of postsynaptic AMPA receptors in response to the lack of *Efnb1*. Of note, 22–24 neurons from 5 to 6 animals were analyzed per group. Data are shown ± SEM. Statistical significance was determined using 2-tailed Student *t* test (C, D). *P ≤ 0.05, ***P ≤ 0.001 versus *Efnb1*[loxP/loxP] male mice (C, D). The underlying data are provided in S1 Data. AMPA, α-amino-3-hydroxy-5-methyl-4-isoxazolepropionic acid; SEM, standard error of the mean; sEPSC, spontaneous excitatory postsynaptic currents.

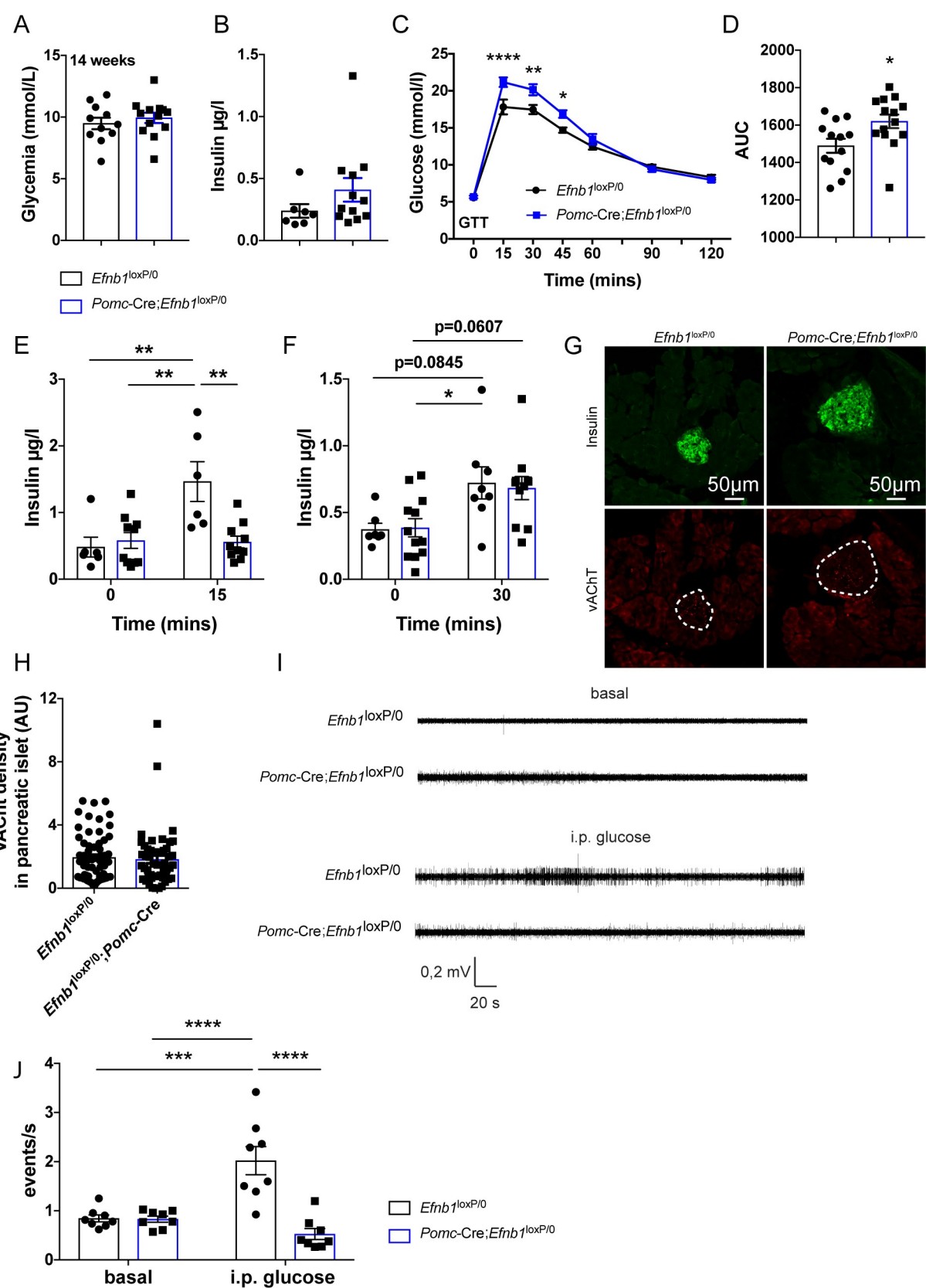

**Fig 6. Loss of *Efnb1* in POMC-expressing progenitors causes glucose intolerance in males.** (A) Basal glycemia of 14-week-old male mice ($n$ = 11–13/group). (B) Basal insulinemia of 16–18-week-old male mice ($n$ = 7–12/group). (C) Glucose tolerance test of 8–9-week-old male mice ($n$ = 13–14/group). (D) Area under the curve of the glucose tolerance test. Glucose-induced insulin secretion test of 9–10-week-old male mice ($n$ = 6-12/group) 15 minutes (E) and 30 minutes (F) after glucose injection. (G) Confocal images illustrating cholinergic vAChT-positive fibers (red) in pancreatic islets labeled with insulin (green) of 16–18-week-old male mice. (H) Quantification of cholinergic fiber density in pancreatic islets ($n$ = 3 animals/group, between 16 and 27 islets per animal). Representative traces (I) and quantification (J) of parasympathetic nerve firing rate in the basal state and following IP glucose injection in mice ($n$ = 8/group). Data are shown ± SEM. Statistical significance was determined using 2-way ANOVA (C, E, F, J) and 2-tailed Student $t$ test (A, B, D, H). $^*P \leq 0.05$ versus $Efnb1^{\text{loxP/0}}$ (C, D) and versus $Pomc$-Cre; $Efnb1^{\text{loxP/0}}$ (F); $^{**}P \leq 0.01$ versus $Efnb1^{\text{loxP/0}}$ (C, E) and versus $Efnb1^{\text{loxP/0}};Pomc$-Cre (E); $^{***}P \leq 0.001$ versus $Efnb1^{\text{loxP/0}}$ (J) and versus $Efnb1^{\text{loxP/0}};Pomc$-Cre (J); $^{****}P \leq 0.0001$ versus $Efnb1^{\text{loxP/0}}$ (C). The underlying data are provided in S1 Data. ANOVA, analysis of variance; GTT, glucose tolerance test; IP, intraperitoneal; POMC, proopiomelanocortin; SEM, standard error of the mean.

6D). After 15 but not 30 minutes, glucose-stimulated insulin secretion was impaired in mutant mice, suggesting that only the cephalic phase of insulin secretion (first phase) was impacted (Fig 6E and 6F). Activation of the cholinergic parasympathetic innervation of the pancreatic islets, and inhibition of the sympathetic nervous system control insulin secretion in response to hyperglycemia [33]. We thus assessed cholinergic (parasympathetic) innervation of pancreatic islets in $Pomc$-Cre;$Efnb1^{\text{loxP/0}}$ and $Efnb1^{\text{loxP/0}}$ male mice (Fig 6G and 6H). No difference in the density of cholinergic fibers was found in islets of $Pomc$-Cre;$Efnb1^{\text{loxP/0}}$ mice compared with those of $Efnb1^{\text{loxP/0}}$ male mice (Fig 6H). We then measured parasympathetic nerve (vagus) activity upon glucose challenge. We observed no difference in the basal firing activity between $Pomc$-Cre;$Efnb1^{\text{loxP/0}}$ male mice and $Efnb1^{\text{loxP/0}}$ control littermates (Fig 6I and 6J). However, whereas a glucose challenge increased firing activity by 2.5-fold in $Efnb1^{\text{loxP/0}}$ control mice, no response was detected in the vagus nerve of $Pomc$-Cre;$Efnb1^{\text{loxP/0}}$ mice (Fig 6J). We also performed pyruvate and insulin tolerance tests, and the results were identical in control and mutant mice (S4 Fig). Notably, similar metabolic disturbances were found in $Pomc$-Cre;$Efnb1^{\text{loxP/loxP}}$ female mice (S4 Fig). Together, these data show that mice lacking $Efnb1$ in POMC$^{\text{prog}}$ display altered excitability of POMC$^{\text{prog}}$ and develop glucose intolerance that is associated with impaired insulin secretion and impaired parasympathetic nerve activity.

In control mice, $Efnb1$ mRNA was not detectable in the adeno-pituitary of late embryos where adrenocorticotropic hormone (ACTH) neurons (derived from POMC precursor) can also be found (S4 Fig); nonetheless, $Efnb1$ mRNA was expressed in adult pituitary (S4 Fig). The deletion of $Efnb1$ in POMC neurons did not affect the expression of $Efnb1$ or $Efnb2$ mRNA in the adult pituitary (S4 Fig). However, we cannot exclude that POMC neurons expressed in the pituitary will not contribute to the phenotype of the mice lacking $Efnb1$ in POMC$^{\text{prog}}$.

## Lack of *Efnb2* in POMC-expressing progenitors impairs feeding and gluconeogenesis in a sex-specific manner

To study the role of $Efnb2$ in the development of glutamatergic terminals in POMC$^{\text{prog}}$, we crossed $Efnb2^{\text{loxP}}$ mice [34] with $Pomc$-Cre mice. As expected, the level of $Efnb2$ mRNA was significantly reduced in the ARH of $Pomc$-Cre;$Efnb2^{\text{loxP/loxP}}$ mice, whereas the level of $Efnb1$ mRNA was unchanged between mutant and control mice (Fig 7A and 7B). $Efnb2$ mRNA was also detected in the adeno-pituitary during late fetal and adult life (S4 Fig). However, no change was observed in $Efnb2$ mRNA expression in the pituitaries of mice that lack $Efnb2$ in their POMC cells when compared to control mice (S5 Fig). Again, we cannot exclude that POMC neurons expressed in the pituitary will not contribute to the phenotype of the mice lacking $Efnb2$ in POMC$^{\text{prog}}$.

The lack of $Efnb2$ in POMC$^{\text{prog}}$ affected glutamatergic inputs into these neurons found in dorsal and lateral parts of the ARH, with a 55% decrease in the number of vGLUT2-positive

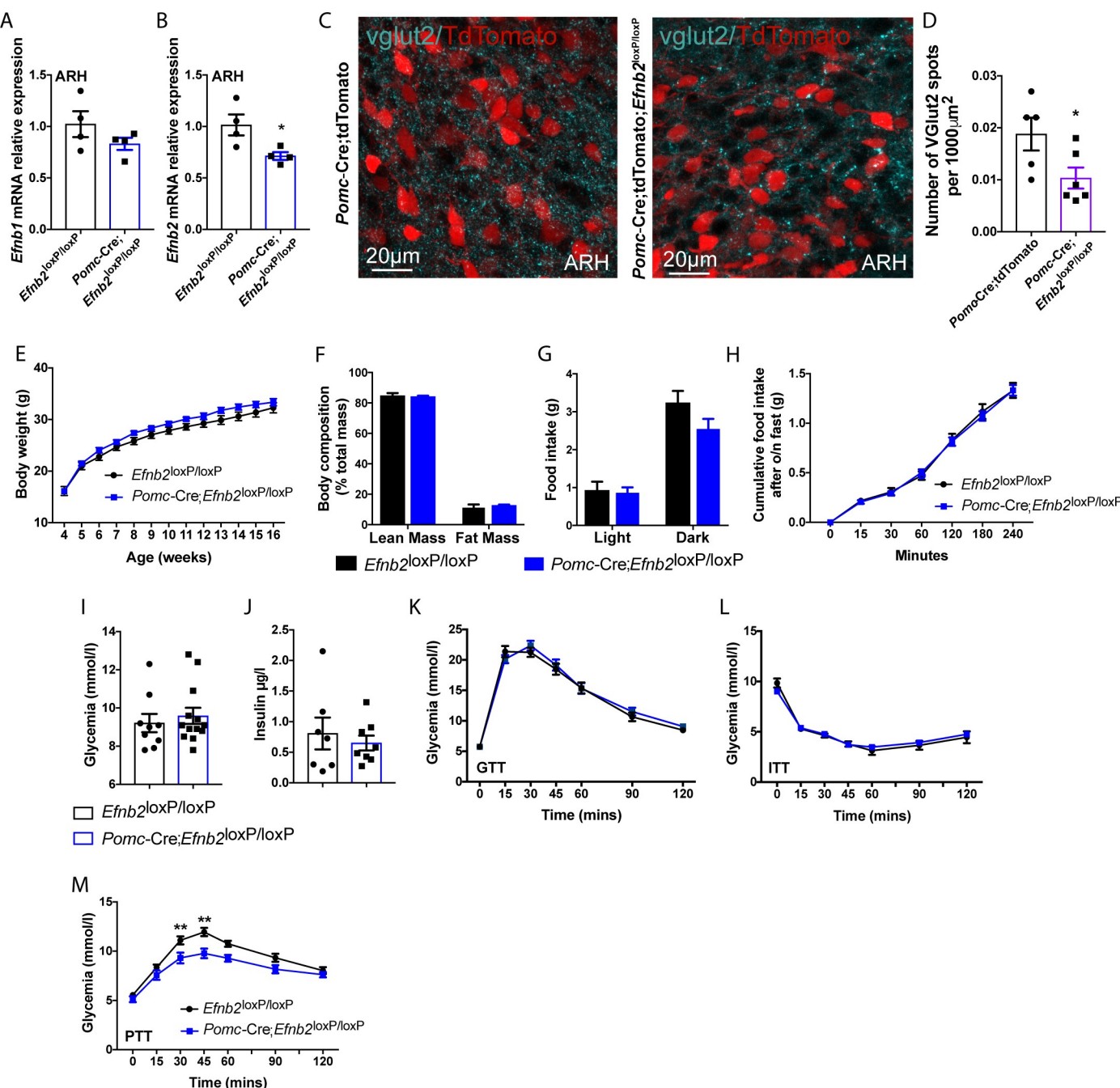

**Fig 7. Loss of *Efnb2* in POMC-expressing progenitors causes impaired gluconeogenesis in males.** *Efnb1* (A) and *Efnb2* (B) mRNA relative expressions in ARH of adult *Efnb2*^loxP/loxP and *Pomc*-Cre;*Efnb2*^loxP/loxP male mice (*n* = 4/group). (C) High magnification of vGLUT2-positive terminals into POMC-expressing progenitors (red) in *Pomc*-Cre;tdTomato control and *Pomc*-Cre;*Efnb2*^loxP/loxP;tdTomato mutant male mice. (D) Quantification of vGLUT2-positive inputs in direct apposition with *Pomc*-Cre;tdTomato neurons (red) in female mice (*n* = 3/group, 2 sections/animal). (E) Post-weaning growth curve of *Efnb2*^loxP/loxP and *Pomc*-Cre;*Efnb2*^loxP/loxP male mice (*n* = 11–14/group). (F) Body composition of 16-week-old male mice (*n* = 7–12/group). (G) Food intake of 13–14-week-old male mice (*n* = 6–10/group). (H) Refeeding after overnight fasting of 13–14-week-old male mice (*n* = 13–15/group). (I) Basal glycemia of 8-week-old male mice (*n* = 9-13/group). (J) Basal insulinemia of 16-week-old male mice (*n* = 7-8/group). (K) Glucose tolerance test of 8–9-week-old male mice (*n* = 11–13/group). (L) Insulin tolerance test of 14-week-old male mice (*n* = 7–12/group). (M) Pyruvate tolerance test of 12–13-week-old male mice (*n* = 11–13/group). Data are shown ± SEM. Statistical significance was determined using 2-way ANOVA (D–G; J–L) and 2-tailed Student *t* test (A, B, D; I, J). $^*P \leq 0.05$ versus *Efnb2*^loxP/loxP (B) and versus *Pomc*-Cre; tdTomato (D); $^{**}P \leq 0.01$ versus *Efnb2*^loxP/loxP (M). The underlying data are provided in S1 Data. ANOVA, analysis of variance; ARH, arcuate nucleus of the hypothalamus; ITT, insulin tolerance test; POMC, proopiomelanocortin; PTT, pyruvate tolerance test; SEM, standard error of the mean.

terminals that were in contact with POMC$^{prog}$ (Fig 7C and 7D). Physiologically, male and female mice lacking *Efnb2* in POMC$^{prog}$ had normal body weight growth curves, body composition, and daily food intake (Fig 7E–7G and S5 Fig). The cumulative food intake in the refeeding paradigm after overnight fasting were only comparable between control and mutant male mice (Fig 7H). In females, cumulative food intake after overnight fasting was significantly increased after 180 and 240 minutes in *Pomc*-Cre;*Efnb2*$^{loxP/loxP}$ mice when compared with *Efnb2*$^{loxP/loxP}$ mice (S5 Fig). In addition, basal glycemia, insulinemia, and glycemia levels after glucose and insulin tolerance test results were similar between the groups of male and female mice (Fig 7I–7L and S5 Fig). Pyruvate tolerance tests indicated impaired gluconeogenesis only in *Pomc*-Cre;*Efnb2*$^{loxP/loxP}$ male mice (Fig 7M and S5 Fig).

Together, these results suggest that the lack of *Efnb2* in POMC$^{prog}$ impairs gluconeogenesis in males and impairs food intake in a refeeding paradigm in females.

## Discussion

Energy and glucose homeostasis are tightly controlled by the brain. POMC and AgRP/NPY neurons in the ARH are key regulators of these functions and respond to peripheral signals through projections to second-order neurons controlling endocrine and autonomic nervous systems. However, POMC and AgRP/NPY neurons also receive extensive inputs from a plethora of areas of the brain [18] and are thus integrated in a complex neuronal network. Although our understanding of the control of feeding behavior and glucose homeostasis has improved over the last few decades, it is still largely unknown how central circuits that regulate POMC activity and their associated functions are being assembled.

Arcuate neuronal populations are very heterogenous, and a common pool of POMC progenitors can give rise to NPY- and Kiss-expressing neurons or remain POMC neurons [28,29]. In adult animals, only a limited proportion of POMC$^{prog}$ expresses POMC [35]. Interestingly, the number of POMC neurons is higher during postnatal ages and decreases over time [12]. Consequently, the proportion of POMC$^{prog}$ that are POMC neurons is more important in pups and particularly at P14 when we performed the cell sorting compared with that described in adults. In this study, we used *Pomc*-Cre mouse model, and conditional deletion using the Cre-Lox system will consequently affect POMC neurons and also NPY and Kiss-expressing cells. We cannot exclude that the phenotype we observed is also caused by the lack of *Efnb1* and *Efnb2* in AgRP/NPY neurons and will be further discussed. However, we focused our histological and electrophysiological analysis on neurons exclusively located in the dorsal and lateral parts of the ARH where most of POMC neurons are found [12].

Here, we showed that there is an enrichment of EphrinB members in POMC neurons when glutamatergic inputs develop, and we described the role of ephrin signaling in the control of the amount of excitatory inputs. EphrinB1 and EphrinB2 appear to both control the number of glutamatergic terminals on POMC neurons; however, they play a distinct role in controlling energy and glucose homeostasis. These are interesting findings given that glutamatergic input pattern is impaired both in mice lacking EphrinB1 and in those lacking EphrinB2 in POMC$^{prog}$, suggesting functional heterogeneity in POMC neuronal circuits.

The present study is in agreement with previous work performed in rats [27] and it shows that in mice, the amount of glutamatergic inputs into arcuate POMC neurons increases gradually after birth until weaning. During this important period of neuronal connectivity formation, we showed that POMC neurons were enriched with EphrinB1 and EphrinB2, 2 members of the ephrin family. These proteins enable cell-to-cell contacts through interaction with EphA and EphB receptors to control axon growth, synaptogenesis, or synaptic plasticity. We focused our study on Ephb1 and Ephb2 receptors because their interactions with EphrinB1 and

EphrinB2 are well described [30], and EphrinB members are known to play a key role in AMPA and NMDA glutamate receptor recruitment, stabilization of glutamatergic synapses, and control of the number of excitatory synapses [22–26].

Here, we used a developmental approach that interfered with excitatory synaptic input formation to specifically assess the role of glutamatergic inputs in POMC[prog] in the control of energy and glucose homeostasis. We showed that lacking EphrinB1 or EphrinB2 in POMC[prog] reduces the amount of glutamatergic input into these neurons. Interestingly, these 2 mouse models do not have similar physiological outcomes, suggesting specificity in the establishment of glutamatergic input patterns. We first hypothesized that these differences arose from POMC heterogeneity, as distinct subsets of leptin receptor-, insulin receptor- and serotonin receptor-expressing POMC neurons are linked to functional differences [36,37]; however, our data showed that every detected POMC neuron expressed both *Efnb1* and *Efnb2*. Thus, the differential effects of *Efnb1* and *Efnb2* inactivation on energy and glucose metabolism could stem from EphrinB favoring the formation of presynaptic inputs arising from distinct areas. Indeed, several Eph receptors can interact with EphrinB1 and EphrinB2 [30] with different affinities [21] and can also be differentially expressed in areas known to project to POMC neurons. These aspects of the study will require further analysis.

In this study, the reduction of the amount of glutamatergic inputs into POMC[prog] is associated with a decrease of the number of *Gria1* and *Grin1* mRNA expression, 2 AMPAR and NMDAR subunits in POMC neurons, respectively. The AMPAR-mediated sEPSC are consequently affected in POMC[prog]. The amplitude and frequency of the AMPAR-dependent sEPSC recorded in POMC[prog] in the dorsal and lateral parts of the ARH, where most of POMC neurons are observed, are consistent to that described in previous studies for POMC-GFP neurons [38,39]. In line with our results, the lack of *Grin1* in POMC neurons leads to the lack of NMDAR-mediated sEPSC [40].

The loss of *Efnb1* in POMC[prog] is associated with alterations in glucose tolerance and losses of parasympathetic nerve activity and insulin secretion. These findings are consistent with previous studies, which reported that affecting either POMC signaling, circuits, or POMC neuron survival leads to impaired glucose homeostasis [10–12]. Surprisingly, loss of *Efnb1* in POMC neurons does not perturb food intake or body weight. Other studies have shown that ablation or inactivation of arcuate POMC neurons [11,41] as well as genetic deficiency in POMC [42,43] cause hyperphagia and obesity. In addition, chemogenetic stimulation of POMC neurons reduces food intake [44] as well as activation of POMC neurons projecting into the PVH [32]. Notably, the aforementioned studies cannot distinguish the effects of these kinds of input from those related to the output to POMC neurons, and only a few studies have focused on the effect of synaptic inputs onto POMC neurons. Indeed, deletion of glutamatergic NMDA receptor subunits GluN2A, GluN2B, and Grin1 (GluN1) in POMC neurons does not lead to changes in glucose homeostasis [45] neither in body weight and food intake [40]. These studies suggest a role of AMPAR in the phenotypes we observed in mice lacking *Efnb1* and *Efnb2* in POMC[prog]. On the other hand, the functional effect of presynaptic inputs to POMC neurons could also be mediated by another NMDA subunit; both cases are in agreement with our data, in that they cannot distinguish which glutamatergic receptors are predominantly involved. In some cases, POMC functions require long or chronic chemogenetic activation [9,11], which could reflect the recruitment of NMDA receptors alongside AMPA receptors, since $Ca^{2+}$ entry through AMPA receptors precedes full activation of NMDA receptors [46].

The loss of *Efnb2* in POMC[prog] impaired gluconeogenesis in males and food intake in females in a refeeding paradigm. These findings are surprising, as chemogenetic activation and inhibition of arcuate POMC neurons repress and increase hepatic glucose production, respectively. Interestingly, a lack of ephrin expression has been shown to induce local

reorganization of glutamatergic synaptic inputs [47]. The lack of EphrinB2 in POMC[prog] could therefore affect the number of excitatory terminals connecting to nearby AgRP/NPY neurons, which are known to suppress hepatic glucose production through insulin action [7,8] and to promote feeding [48].

Our findings also suggest sexual dimorphism in the melanocortin system, as the lack of *Efnb2* in POMC neurons does not lead to impaired gluconeogenesis in females, but it does result in changes to refeeding after an overnight fast. The ARH has not been primarily viewed as a dimorphic structure, but recent studies showed differences between males and females in the number of ARH POMC neurons, their firing rate, the development of diet-induced obesity, and the activation of STAT3 in POMC neurons [49–51].

The phenotypes we observed in mice lacking *Efnb1* or *Efnb2* can also be due to synaptic plasticity, as cells with EphrinB have been shown to control this function [25] and are found in the brains of adult animals (Allen Mouse Brain Atlas [52]). Moreover, hormones [8,17], metabolic status, physical activity [39], and the age [53] can directly modulate the amount of glutamatergic and GABAergic synaptic inputs into POMC neurons.

In conclusion, our data show that distinct Ephrin members control the glutamatergic innervation of POMC[prog] and specific functions such as glucose homeostasis or feeding. This supports the idea that POMC neuronal network is heterogeneous and that POMC neurons should not be considered as first-order neurons but have to be thought predominantly as integrators of multiple kinds of complex peripheral and central information to control energy and glucose homeostasis.

## Materials and methods

### Experimental model and subject details

**Ethics statement.**   All procedures were conducted in accordance to the Swiss National Institutional Guidelines of Animal Experimentation (OExA; 455.163) with license approval (VD3193) issued by the Cantonal Veterinary Authorities (Vaud, Switzerland).

**Animals.**   Mice were group housed in individual cages and maintained in a temperature-controlled room with a 12-h light/dark cycle and provided ad libitum access to water and standard laboratory chow (Kliba Nafag, Kaiseraugst, Switzerland). Mice were single housed only for food intake experiments. All mice used in this study have been previously described: *Pomc*-Cre [32], ROSA-tdTomato reporter [54], *Efnb1*[loxP/loxP] [31], *Efnb2* [loxP/loxP] [34], *Pomc*-eGFP [13], and *Npy*-hrGFP [55]. *Pomc*-Cre mice were mated to *Efnb1*[loxP/loxP], *Efnb2* [loxP/loxP] to generate *Pomc*-specific *Efnb1* or *Efnb2* knockout mice. As *Efnb1* gene is carried by X chromosome, males have thus only 1 copy (*Efnb1*[loxP/0]).

### Method details

**Monosynaptic retrograde tracing.**   Virus: pAAV-syn-FLEX-splitTVA-EGFP-tTA (Addgene viral prep # 100798-AAV1; http://n2t.net/addgene:100798; RRID:Addgene_100798) and pAAV-TREtight-mTagBFP2-B19G were a gift from Ian Wickersham (Addgene viral prep # 100799-AAV1; http://n2t.net/addgene:100799; RRID:Addgene_100799).

Surgery: Monosynaptic retrograde tracing using rabies virus was performed as follows: Adult Pomc-Cre mice were anesthetized with a mix of xylazine ketamine. Viruses were injected with a microsyringe (Hamilton, 35 G) and microinjection pump (World Precisions Instruments, Sarasota, United States of America, rate at 100 nl/min). Mice receive 300 nl of mixed AAV1-Syn-FLEX-splitTVA-eGFP-tTA and AAV1-TREtight-BFP2-B19G in 1 side of the ARH (AP: −1.4 mm; ML: −0.3 mm; DV: −5.8 mm). After 7 days, the same mice received a second injection of 300 nl of pseudotyped rabies virus EnvA-SADdG-mcherry (Salk Institute)

using the same coordinates. Control mice were injected with helper virus or EnvA-SADdG-mcherry alone. One week later, mice were anesthetized and perfused with 4% PFA and frozen for brain cryosectioning.

## Cell sorting

P14 *Pomc*-Cre;tdTomato;*Npy*-hrGFP mice (*n* = 3 for RNA-seq; *n* = 4 to 8 for RT-qPCR) were microdissected under binocular loupe and enzymatically dissociated using the Papain Dissociation System (Worthington Biochemical Corporation, Lakewood, USA) following the manufacturer's instructions. Fluorescence-activated cell sorting (FACS) was performed using a BD FACS Aria II Cell Sorter to sort *Pomc*-tdTomato⁺, *Npy*-hrGFP⁺ and cells containing both *Pomc*-tdTomato and *Npy*-hrGFP. Non-fluorescent cells obtained from wild-type (WT) animals were used to set the gate of sorting.

## Nucleus microdissection

As already described [56], PVH and ARH nuclei were microdissected under binocular loupe from 200-μm thick brain sections collected from P8, P10, P12, P14, P16, and P18 C57Bl/6 pups (*n* = 2 to 4/age, from at least 2 litters/age) and from 16-week-old *Pomc*-Cre;*Efnb2*$^{loxP/loxP}$ and *Efnb2*$^{loxP/loxP}$. Microdissected nuclei were stored at −80˚C until RNA extraction using Picopure RNA extraction kit (Applied Biosystems, Thermo Fisher Scientific, Waltham, USA).

## RNA sequencing

RNAs were extracted from each sorted cell population using a Picopure RNA extraction kit (Thermo Fisher Scientific, Waltham, USA). RNA integrity and concentration were assessed with a 2100 Bioanalyser (Agilent, Santa Clara, USA). Five hundred picograms of RNAs were reverse transcript using SMART-Seq v4 Ultra Low Input RNA (Takara, Shiga Japan), and RNA-seq libraries were prepared by the Illumina Nextera XT DNA Library kit (Illumina, San Diego, USA). A sequencing depth of 98 to 138 million of reads was used per library. Purity-filtered reads were adapters and quality trimmed with Cutadapt (v. 1.8, Martin 2011). Reads matching to ribosomal RNA sequences were removed with fastq_screen (v. 0.11.1). Remaining reads were further filtered for low complexity with reaper (v. 15–065) [57]. Reads were aligned against Mus musculus.GRCm38.92 genome using STAR (v. 2.5.3a) [57]. The number of read counts per gene locus was summarized with htseq-count (v. 0.9.1) [58] using Mus musculus. GRCm38.92 gene annotation. The quality of the RNA-seq data alignment was assessed using RSeQC (v. 2.3.7) [59]. Reads were also aligned to the Mus musculus.GRCm38.92 transcriptome using STAR [57], and the estimation of the isoform abundance was computed using RSEM (v. 1.2.31) [60]. Statistical analysis was performed for genes in R (R version 3.4.3). Genes with low counts were filtered out according to the rule of 1 count per million (CPM) in at least 3 samples. Library sizes were scaled using TMM normalization and log-transformed into CPM (EdgeR package version 3.20.8) [61]. Moderated *t* test was used for each contrast. The adjusted *P* value is computed by the Benjamini–Hochberg method, controlling for false discovery rate (FDR or adjusted *P* value). These data are available with accession number GSE144887.

## RT-qPCR

For gene expression analyses, cDNA was generated with the high-capacity cDNA Reverse Transcription kit (Applied Biosystems Thermo Fisher Scientific, Waltham, USA). RT was performed on 70 ng of RNAs for microdissected nuclei study on pups, and on 190 and 800 ng,

respectively, for ARH and pituitary studies in adults. RT-qPCR was performed using SyBR green mix (Applied Biosystems Thermo Fisher Scientific, Waltham, USA) and SyBR green primers (Microsynth, Balgach, Switzerland) for *Efnb1*, *Efnb2*, *Ephb1*, *Ephb2*, *Ephb3*, *Ephb4*, *Epha4*, *and Epha5*. Gapdh was used for endogenous control. Primer sequences are as follows:

*Efnb1 rev*: ACAGTCTCATGCTTGCCGTC; *Efnb1 fwd*: CACCCGA GCAGTTGACTACC; *Efnb2 rev*: CCTTGTCCGGGTAGAAATTTGG; *Efnb2 fwd*: GGTTTTGTGCAGAACTGCG AT; *Ephb1 rev*: CTGATGGCCTGCCAAGGTTA; *Ephb1 fwd*: CAGGCTTCACCTCCCTTC AG; *Ephb2 rev*: CTCAAACCCCCGTCTGTTACAT; *Ephb2 fwd*: CTACCCCTCATCGTTG GCTC; *Ephb3 rev*: GAGCTGAGTGTCA GACCTGC; *Ephb3 fwd*: GACTGCAGAAGATCT GCTAAGGA; *Ephb4 rev*: TCTGCGCCCTTCTCATGATACT; *Ephb4 fwd*: TTTCTTTC TC CTGCAGTGCCT; *Epha4 rev*: AGTTCGCAGCCAGTTGTTCT; *Epha4 fwd*: CTGGAAGG AGGGTGGGAGG; *Epha5 rev*: ATTCCATTGGGGCGATCTGG; *Epha5 fwd*: GGTACCTG CCAAGCTCCTTC; *Pomc rev*: TCCAGCGAGAGGTCGAGTTT; *Pomc* fwd: ATGCCGAG ATTCTGCTACAGT; *Npy* rev: CAGCCAGAATGCCCAAACAC; *Npy* fwd: CCGCCACGA TGCTAGGTAAC;*Gapdh rev*: AAGATGGTGATGGGCTT CCC; *Gapdh fwd*: CTCCACTC ACGGCAAATTCA.

All assays were performed using an Applied Biosystems 7500 Fast Real-Time PCR system. Calculations were performed by comparative method ($2^{-\Delta\Delta CT}$).

## In situ hybridization (RNAscope)

P14 male C57Bl/6, *Pomc*-eGFP, and 14 (for monosynaptic retrograde tracing) and 18–19-weeks-old *Pomc*-Cre male mice were perfused with 4% PFA. WT embryos at embryonic day E17 were collected and immerged in 4% PFA overnight. On 20-μm thick brain or embryo coronal cryosections, in situ hybridization for *Efnb1* (cat # 526761, cat # 526761-C2), *Efnb2* (cat # 477671), *Ephb1* (cat # 567571-C3), *Ephb2* (cat # 447611-C3), *Slc17a6 (vglut2)* (cat # 319171), *Gria1* (cat # 426241-C1), and *Grin1* (cat # 431611-C3) was processed using RNAscope probes and RNAscope Fluorescent Multiplex Detection Reagents (Advanced Cell Diagnostics, Newark, USA) following manufacturer's instructions.

## Immunohistochemistry

P6, P14, and P22 *Pomc*-Cre;tdTomato male mice (*n* = 2 to 3 animals/age), 16- to 18-week-old *Pomc*-Cre;tdTomato;*Efnb1*loxP/0, *Pomc*-Cre;tdTomato, *Pomc*-Cre;tdTomato;*Efnb2*loxP/loxP, and *Pomc*-Cre;tdTomato;*Efnb1*loxP/loxP, *Pomc*-Cre;tdTomato;*Efnb2*loxP/loxP, *Pomc*-Cre;tdTomato female mice were transcardially perfused with 4% PFA (*n* = 3/group). Brain and pancreas sections that were 20-μm thick were processed for immunofluorescence using standard procedures [12,56,62]. The primary antibodies used for immunohistochemistry (IHC) were as follows: rabbit anti-vGLUT2 (1:500, Synaptic Systems), rabbit anti-VAChT (1:500, Synaptic Systems, Goettingen, Germany), and guinea pig anti-insulin (1:500, Abcam, Cambridge, United Kingdom). Primary antibody was visualized with Alexa anti-rabbit 647 and 568 and Alexa anti-guinea pig 488 (Thermo Fisher Scientific, Waltham, USA).

## Image analyses

To quantitatively analyze cholinergic (VAChT-positive) fibers in pancreatic cells, between 16 and 27 pancreatic islets per animal were imaged using a LSM 710 (Zeiss, Jena, Germany) confocal system equipped with a ×20 objective. Each image was binarized to isolate labeled fibers from the background and to compensate for differences in fluorescence intensity. The integrated intensity, which reflects the total number of pixels in the binarized image, was then

calculated for each islet. This procedure was conducted for each image. Image analysis was performed using ImageJ analysis software (NIH).

To quantitatively analyze glutamatergic innervation of POMC neurons, adjacent image planes were collected in the lateral part of the ARH through the *z*-axis using a Zeiss LSM 710 confocal system at a frequency of 0.25 μm through the entire thickness of the ARH. Three-dimensional reconstructions of the image volumes were then prepared using Imaris 9.3.1 visualization software. The number of glutamatergic inputs into POMC neurons was quantified. Each putative glutamatergic input was defined as a spot, and we quantified the number of glutamatergic spots that contacted *Pomc*-Cre;tdtomato+.

*Gria1* and *Grin1* mRNA spots in arcuate *Pomc*-expressing neurons of 19- to 20-week-old *Efnb1*$^{loxP}$ and *Pomc*-Cre;*Efnb1*$^{loxP/0}$ male mice were manually quantified using ImageJ software. Two to 3 animals were used per group and mRNA spots were quantified on 206 to 263 neurons.

## Physiological measures

*Pomc*-Cre;tdTomato;*Efnb1*$^{loxP/0}$, *Efnb1*$^{loxP/0}$ ($n$ = 14 to 16 per group), *Pomc*-Cre;tdTomato;*Efnb2*$^{loxP/loxP}$, *Efnb2*$^{loxP/loxP}$ ($n$ = 11 to 14 per group) male mice and *Pomc*-Cre;tdTomato;*Efnb1*$^{loxP/loxP}$, *Efnb1*$^{loxP/loxP}$ ($n$ = 8 to 10 per group), *Pomc*-Cre;tdTomato;*Efnb2*$^{loxP/loxP}$, *Efnb2*$^{loxP/loxP}$ ($n$ = 19 per group) female mice were weighed every week from 3 weeks (weaning) to 16 weeks using an analytical balance. To measure food consumption, 13- to 14-week-old *Pomc*-Cre;tdTomato;*Efnb1*$^{loxP/0}$, *Efnb1*$^{loxP/0}$ ($n$ = 9 to 12 per group), *Pomc*-Cre;tdTomato;*Efnb2*$^{loxP/loxP}$, *Efnb2*$^{loxP/loxP}$ ($n$ = 6 to 10 per group) male mice and *Pomc*-Cre;tdTomato;*Efnb2*$^{loxP/loxP}$, *Efnb2*$^{loxP/loxP}$ ($n$ = 8 per group) female mice were housed individually in BioDAQ cages (Research Diets, New Brunswick, USA), and after at least 2 days of acclimation, food intake was assessed on 2 consecutive days. The means obtained on these 2 days were used for analyses. Body composition analysis (fat/lean mass) was performed in 16-week-old *Pomc*-Cre;tdTomato;*Efnb1*$^{loxP/0}$, *Efnb1*$^{loxP/0}$ ($n$ = 9 to 10 per group), *Pomc*-Cre;tdTomato;*Efnb2*$^{loxP/loxP}$, *Efnb2*$^{loxP/loxP}$ ($n$ = 7 to 12 per group) male mice and *Pomc*-Cre;tdTomato;*Efnb2*$^{loxP/loxP}$, *Efnb2*$^{loxP/loxP}$ ($n$ = 6 per group) female mice using nuclear magnetic resonance (NMR; Echo MRI). Glucose (GTT), insulin (ITT) and pyruvate (PTT) tolerance tests were conducted in 8- to 12-week-old *Pomc*-Cre;tdTomato;*Efnb1*$^{loxP/0}$, *Efnb1*$^{loxP/0}$ ($n$ = 9 to 14 per group), *Pomc*-Cre;tdTomato;*Efnb2*$^{loxP/loxP}$, *Efnb2*$^{loxP/loxP}$ ($n$ = 8 to 13 per group) male mice and *Pomc*-Cre;tdTomato;*Efnb2*$^{loxP/loxP}$, *Efnb2*$^{loxP/loxP}$ ($n$ = 8 to 14 per group) female mice through intraperitoneal (IP) injection of glucose (2 mg/g body weight), insulin (0.5 U/kg body weight) or sodium pyruvate (2 mg/g body weight) after overnight fasting (15 to 16 h, GTT and PTT) or 5 to 6 h of fasting (ITT). Blood glucose levels were measured at 0, 15, 30, 45, 60, 90, and 120 min postinjection. Glycemia was measured using a glucometer (Bayer, Leverkusen, Germany).

Glucose-stimulated insulin secretion tests were also performed at 9 to10 weeks of age, through the IP administration of glucose (2 mg/kg body weight, $n$ = 6 to 12 per group) after 15 to 16 h overnight fasting. Blood samples were collected 0 and 15 min and 0 and 30 min after glucose injection on 2 distinct cohorts. Serum insulin levels were then measured using an insulin ELISA kit (Mercodia, Uppsala, Sweden). Basal insulinemia was measured on 16 to 18-week-old mice ($n$ = 6 to 12/group) using an insulin ELISA kit (Mercodia). Basal glycemia was measured the morning on fed mice using a glucometer (Bayer, Leverkusen, Germany).

## Vagus nerve activity recording

The firing rate of the thoracic branch of the vagal nerve along the carotid artery was recorded as previously described [63–65] on 14 to 15-week-old *Pomc*-Cre;tdTomato;*Efnb1*$^{loxP/0}$,

*Efnb1*^loxP/0^ (*n* = 8/group). Unipolar nerve activity was recorded continuously under isoflurane anesthesia (30 min during basal condition and 30 min after IP glucose at a dose of 2 g/kg) using the LabChart 8 software (AD Instrument, Oxford, United Kingdom). Data were digitized with PowerLab 16/35 (AD Instrument). Signals were amplified $10^5$ times and filtered using 200/1,000 Hz band pass filter. Firing rate analysis was performed using LabChart 8. Data were analyzed for 6 min at the end of the basal recording and for the same duration 15 min after glucose IP injection.

## Electrophysiology recording

Four- to 6-week-old male and female mice were deeply anesthetized with isoflurane before decapitation, and coronal sections (250-μm thick) containing ARH were prepared using a vibratome (VT1000S, Leica, Wetzlar, Germany) and maintained at controlled temperature (32˚C) and oxygenation for at least 1 h before recording. Experiments were performed on a BX51WI upright microscope (Olympus, Tokyo, Japan) mounted on a motorized stage and coupled to micromanipulators (Sutter Instruments, Novato, USA). Slices were placed in a submerged-type recording chamber under the microscope (JG-23W/HP, Warner Instruments, Hamden, USA) and continuously superfused at a flow rate of 2 ml/min with oxygenated ACSF solution maintained at 32 to 34˚C and containing (in millimolars): 126 NaCl, 1.6 KCl, 1.1 $NaH_2PO_4$, 1.4 $MgCl_2$, 2.4 $CaCl_2$, 26 $NaHCO_3$, and 11 glucose (295 to 305 mOsm). tdTomato-positive POMC-expressing progenitors located in the dorsal and lateral parts of the ARH were identified using adequate light excitation delivered by a mercury lamp (U-LH100HG, Olympus) and fluorescence filters. Borosilicate glass pipettes (Harvard Apparatus, Holliston, USA) with tip resistances ranging from 3 to 7 MΩ were shaped with a horizontal micropipette puller (P-97, Sutter Instruments) and used to obtain whole-cell recordings from visually identified neurons. The intrapipette solution contained (in millimolars): 117 cesium methansulfonate, 20 HEPES, 0.4 EGTA, 2.8 NaCl, 5 TEA-Cl, 2.5 MgATP, and 0.25 NaGTP (pH 7.2–7.3; 275 ± 5 mOsm). Whole-cell recordings were performed using a MultiClamp 700B amplifier associated with a 1440A Digidata digitizer (Molecular Devices, San Jose, USA). Neurons with an access resistance exceeding 25 MΩ or changed by more than 20% during the recording were excluded. Bridge balance and pipette capacitance were adjusted before recording. Neurons were voltage clamped at −70 mV in the presence of picrotoxin (100 μM) in order to block $GABA_A$ receptor-mediated inhibitory postsynaptic currents and to isolate spontaneous AMPAR-mediated excitatory postsynaptic currents (sEPSC). EPSC were filtered at 2 kHz, digitized at 10 kHz, and collected online using pClamp 10 (Molecular Devices, San Jose, USA). Quantitative analysis of sEPSC was performed using the Mini Analysis software (Synaptosoft, Decatur, USA) on 22 to 24 neurons per group from 5 to 6 animals.

## Quantification and statistical analysis

All values were represented as the mean ± standard error of the mean (SEM). Statistical analyses were conducted using GraphPad Prism (version 7). Statistical significance was determined using unpaired 2-tailed Student *t* test, 1-way analysis of variance (ANOVA) followed by Tukey post hoc test, and 2-way ANOVA followed by Sidak post hoc test when appropriate. $P \leq 0.05$ was considered statistically significant.

## Supporting information

**S1 Fig. *Efnb1* and *Efnb2* mRNA are enriched in POMC neurons.** (A) PCA made on 14,607 genes. (B) Scatterplots comparing the expression of individual genes between Pomc->Pomc and Npy->Npy neuronal population at P14. (C) Microphotographs showing *Efnb1* (blue) and

*Efnb2* (red) mRNA spots in POMC-GFP$^+$ (green) neurons in the ARH of P14 male mice. (D) High magnification of the inset shown in (C). Quantification of the number of *Efnb1* mRNA spots (E), *Efnb2* mRNA spots (F) or *Efnb1*/*Efnb2* mRNA spot ratio (G) in POMC-eGFP neurons in the entire thickness of the ARH of P14 male animals ($n$ = 2 animals). (H) Schematic illustrating the subdivisions of the ARH used for the quantification in E, F, and G. Data are shown ± SEM. Statistical significance was determined using 1-way ANOVA (E–G). The underlying data are provided in S1 Data. ANOVA, analysis of variance; ARH, arcuate nucleus of the hypothalamus; DMH, dorsomedial nucleus of the hypothalamus; GFP, green fluorescent protein; NPY, neuropeptide Y; PCA, principal component analysis; POMC, proopiomelanocortin; RCH, retrochiasmatic area; SEM, standard error of the mean; VMH, ventromedial nucleus of the hypothalamus; V3, third ventricle.
(TIF)

**S2 Fig. Glutamatergic PVH neurons project into POMC neurons of the ARH.** (A) Experimental approach. A mix of AAV-TREtight-mTagBFP2-B19G and AAV-syn-FLEX-splitT-VA-EGFP-tTA was injected at day 0 in the ARH of 12-week-old *Pomc*-Cre male mice. Seven days later, mice received injection of EnvA-G-deleted-mcherry pseudotyped rabies. Animals were perfused 1 week later for further analyses. (B) Photomicrographs showing the co-localization of mcherry-positive cells (POMC inputs) with glutamatergic neurons of the PVH (*vglut2* mRNA spots in green). DAPI counterstaining is shown in blue. ARH, arcuate nucleus of the hypothalamus; POMC, proopiomelanocortin; PVH, paraventricular nucleus of the hypothalamus; V3, third ventricle.
(TIF)

**S3 Fig. Eph receptors are expressed in PVH during postnatal development.** Quantification of *Ephb3* (A), *Ephb4* (B), *Epha4* (C) and *Epha5* (D) mRNA relative expression in PVH of P8, P10, P12, P14, P16 and P18 male pups ($n$ = 2–4 pups/age). Data are shown ± SEM. Statistical significance was determined using 1-way ANOVA (A–D). $^*P \leq 0.05$ versus P12 (A), versus P14 (A), $^{**}P \leq 0.01$ versus P8 (A), versus P14 (D). The underlying data are provided in S1 Data. ANOVA, analysis of variance; PVH, paraventricular nucleus of the hypothalamus; SEM, standard error of the mean.
(TIF)

**S4 Fig. Loss of *Efnb1* in POMC-expressing progenitors causes glucose intolerance in females.** (A) Microscope image illustrating the expression of *Efnb1* (blue) and *Efnb2* (red) mRNA in the adeno-pituitary of E17 embryo. DAPI counterstaining is shown in white. (B) *Efnb1* and *Efnb2* mRNA relative expression in the pituitary of *Efnb1*$^{loxP/0}$ and *Pomc*-Cre;*Efnb1*$^{loxP/0}$ 16-week-old male mice ($n$ = 3–5/group). (D) Insulin tolerance test of 14-week-old male mice ($n$ = 10–13/group). (E) Pyruvate tolerance test of 13-week-old male mice ($n$ = 9–10/group). (F) Post-weaning growth curve of *Efnb1*$^{loxP/loxP}$ and *Pomc*-Cre;*Efnb1*$^{loxP/loxP}$ female mice ($n$ = 8–11/group). (G) Basal glycemia of 8-week-old female mice ($n$ = 7–11/group). (H) Glucose tolerance test of 8–9-week-old female mice ($n$ = 9/group). (I) Area under the curve of GTT experiment. (J) Insulin tolerance test of 14-week-old female mice ($n$ = 9/group). (K) Pyruvate tolerance test of 13-week-old female mice ($n$ = 7–8/group). Data are shown ± SEM. Statistical significance was determined using 2-way ANOVA (D–F, H, J, K) and 2-tailed Student $t$ test (B, C, G, I). $^{***}P \leq 0.001$ versus *Efnb1*$^{loxP/loxP}$ (H). The underlying data are provided in S1 Data. ANOVA, analysis of variance; GTT, glucose tolerance test; ITT, insulin tolerance test; POMC, proopiomelanocortin; PTT, pyruvate tolerance test; SEM, standard error of the mean.
(TIF)

**S5 Fig. Loss of *Efnb2* in POMC-expressing progenitors causes impaired refeeding after overnight fast in females.** (A) *Efnb2* mRNA relative expression in the pituitary of *Efnb2*$^{loxP/loxP}$ and *Pomc*-Cre;*Efnb2*$^{loxP/loxP}$ male mice ($n = 4$–6/group). (B) Post-weaning growth curve of *Efnb2*$^{loxP/loxP}$ and *Pomc*-Cre;*Efnb2*$^{loxP/loxP}$ female mice ($n = 19$/group). (C) Body composition of 16-week-old female mice ($n = 6$/group). (D) Food intake of 13–14-week-old female mice ($n = 8$/group). (E) Refeeding after overnight fasting of 13–14-week-old female mice ($n = 11$–15/group). (F) Basal glycemia of 8-week-old female mice ($n = 11$–12/group). (G) Basal insulinemia of 16-week-old female mice ($n = 6$–8/group). (H) Glucose tolerance test of 8–10-week-old female mice ($n = 13$–14/group). (I) Insulin tolerance test of 14-week-old female mice ($n = 8$–10/group). (J) Pyruvate tolerance test of 12–13-week-old female mice ($n = 8$–14/group). Data are shown ± SEM. Statistical significance was determined using 2-way ANOVA (B–E, H–J) and 2-tailed Student $t$ test (A, F, G). $^*P \leq 0.05$ versus *Efnb2*$^{loxP/loxP}$ (E); $^{**}P \leq 0.01$ versus *Efnb2*$^{loxP/loxP}$ (E). The underlying data are provided in S1 Data. ANOVA, analysis of variance; POMC, proopiomelanocortin; SEM, standard error of the mean.
(TIF)

**S1 Table. List of putative genes involved in synapse formation and axon guidance.**
(DOCX)

**S1 Data. Original data for the graphs in Figs 1–7 and S1 and S3–S5 Figs.** Each tab includes data for the noted panels in Figs 1–7 and S1 and S3–S5 Figs.
(XLSX)

## Acknowledgments

We thank the CIG Genomic Technologies Facility (GTF) for RNA sequencing experiments and analyses, the EPFL Flow Cytometry Core, the UNIL Flow Cytometry Facilities for cell sorting, and the CIG Animal Facility for their assistance with animal husbandry. We thank Dr. A. Davy (Toulouse, France) for kindly providing *Efnb1*$^{loxP/loxP}$ and *Efnb2*$^{loxP/loxP}$ mice. We are also grateful to Marc Lanzillo with his assistance with animal genotyping.

## Author Contributions

**Conceptualization:** Sophie Croizier.

**Formal analysis:** Gwenaël Labouèbe, Alexandre Picard, Sophie Croizier.

**Funding acquisition:** Sophie Croizier.

**Methodology:** Manon Gervais, Gwenaël Labouèbe, Alexandre Picard, Sophie Croizier.

**Project administration:** Sophie Croizier.

**Supervision:** Sophie Croizier.

**Writing – original draft:** Sophie Croizier.

**Writing – review & editing:** Gwenaël Labouèbe, Alexandre Picard, Bernard Thorens, Sophie Croizier.

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
