## [Editor Report · Decision Letter 0]

3 Feb 2020

Dear Dr Croizier, 

Thank you for submitting your manuscript entitled "EphrinB1 modulates glutamatergic inputs into POMC neurons and controls glucose homeostasis" for consideration as a Research Article by PLOS Biology.

Your manuscript has now been evaluated by the PLOS Biology editorial staff, as well as by an Academic Editor with relevant expertise, and I am writing to let you know that we would like to send your submission out for external peer review.

Please re-submit your manuscript within two working days, i.e. by Feb 05 2020 11:59PM.

Kind regards,

Gabriel Gasque, Ph.D.,

Senior Editor

PLOS Biology

---

## [Decision Letter · Decision Letter 1]

4 Mar 2020

Dear Dr Croizier,

Thank you very much for submitting your manuscript "EphrinB1 modulates glutamatergic inputs into POMC neurons and controls glucose homeostasis" for consideration as a Research Article at PLOS Biology. Your manuscript has been evaluated by the PLOS Biology editors, by an Academic Editor with relevant expertise, and by three independent reviewers.

In light of the reviews (below), we will not be able to accept the current version of the manuscript, but we would welcome re-submission of a much-revised version that takes into account the reviewers' comments. We cannot make any decision about publication until we have seen the revised manuscript and your response to the reviewers' comments. Your revised manuscript is also likely to be sent for further evaluation by the reviewers.

We expect to receive your revised manuscript within 2 months. 

**IMPORTANT - SUBMITTING YOUR REVISION**

Your revisions should address the specific points made by each reviewer, with additional data where the reviewers have indicated. Please submit the following files along with your revised manuscript:

*Re-submission Checklist*

*Published Peer Review*

*PLOS Data Policy*

*Blot and Gel Data Policy*

Sincerely,

Gabriel Gasque, Ph.D., 

Senior Editor

PLOS Biology

REVIEWS:

Reviewer #1: In this manuscript, the authors attempt to assign a role of Ephrin B1 and B2 in the development of glutamatergic Pomc neurons of the arcuate hypothalamus. They provide evidence that glut +ve neurons of the paraventricular hypothalamus express ephrin receptors that may interact with Ephrin B1/2 of Pomc neurons to stabilize Glut termini and promote formation of glut termini by Pomc neurons. The authors also provide evidence that ablation of Ephrin B1 or B2 using the Pomc-Cre allele results in decreased glut termini of Pomc neurons, and has implications in glucose homeostasis (in the case of B1), and glucose homeostasis or impaired food intake that are sex specific. This is generally a well-written manuscript. There are, however, noticeable technical weaknesses, as well as weakness in correctly interpreting results. The PVH story is loosely connected, as the authors do not show localization of ephrin receptors in PVH neurons, and do not knock out the receptors to show the downstream effect on Pomc neurons. Some of the staining is not convincing that makes conclusions questionable. In detail:

Line 75: Citations are missing.

Lines 70-76: Paragraph ends abruptly. Is text missing? Either way, this part should be re-written.

Lines 104-107: In the principal component analysis illustrated in Fig S1, it appears that NPY->NPY cells are vastly different from POMC->NPY cells, as p14-33NPY->NPY appears as an outlier. This is opposite to what authors claim. Regardless, FACS adds considerable manipulations and lengthy ex vivo incubation, hence making transcriptional profiling non-physiological. What was the viability percentage of cells after papain dissociation? If high, then it also poses as a confound regarding expression profiling and cell type survival. Do the FACS proportion of NPY->NPY, POMC->NPY and POMC->POMC cells recapitulate in vivo observations?

Lines 112-113: Spell out what cells were enriched for Efnb1/2.

Lines 116-122: Pomc-GFP +ve neurons and Pomc-Cre-tdTomato neurons minimally overlap (Padilla et al., Endocrinology 2012). Authors cannot compare distribution of Efnb1/2 in Pomc-GFP neurons with FACS/RNAseq data generated with the Pomc-Cre-tdTomato mouse.

Line 133: typo

Line 138: It is not very convincing that the mCherry +ve cells are also glutamatergic. Cells at the top left of the 50um image in Fig. S2B seem Vglut2 +ve but they are not mCherry positive. Moreover, the quality of the tissue seems compromised.

Fig. 2A: Is this RT_PCR from total RNA from the whole PVH? Specify. What is the expression of these genes before or after glutamatergic terminals are formed?

Fig. 2B: Image resolution is poor but it looks like Vglut2 mRNA staining is more convincing than in Figure S2B. How so? Images in Figure 1, Figure 2, Figure 4, Figure S1, S2, S4 require nuclear staining. 

Fig.2D: what is depicted in yellow and what is in blue? Not clearly labelled

Fig. 3: Method by which ARH/PVH cells were co-cultured in vitro is not described. Again, Pomc-eGFP and Pomc-Cre-tdtomato lines cannot be interchangeably used because they label virtually different cells.

Lines 160-162: Did the authors look at localization of the receptors in PVH neuro termini? Why not knock down EphB1/B2 and look at Pomc glut. termini?

Fig. 4: DAPI is missing.

Lines 174-176 and 216-218: Weak conclusions. The measurement of Efnb1/2 mRNA in the whole pituitary may have diluted out the lack of mRNA in the Pomc cells of the pituitary. Just because total pituitary expression is not affected, does not mean that the Pomc cells of the pituitary are not contributing to the phenotype.

Fig.6. Where are the images of glut termini on Pomc neurons?

Line 225: typo

Reviewer #2: The manuscript by Gervais and colleagues examines the role of EphrinB1 in Pomc neurons in the regulation of glucose homeostasis. Here, they report an enrichment of EphrinB members in Pomc neurons when glutamatergic inputs develop. While mice lacking Efnb1 in POMC neurons exhibit impaired glucose tolerance, in part through decreased glucose-induced insulin secretion these effects were not observed in mice lacking Efnb2 in Pomc neurons. Taken together, the authors conclude that ephrins control excitatory input to Pomc neurons to regulate glucose homeostasis. Overall, the studies are well-designed and conducted and the data are of interest.

The primary finding is that mice lacking Efnb1 in POMC neurons exhibit impaired glucose tolerance, an effect associated with reduced glucose-induced insulin secretion. The authors suggest that this is due to blunted vagus nerve activity. Is pharmacological activation of the PNS sufficient to rescue the effect or does blockade of the PNS worsen glucose tolerance in WT mice, but not in KO mice?

The current studies were conducted only in mice fed standard chow and both the energy- and glucose-homeostasis phenotype may be further exaggerated on animals subjected to a high-fat diet.

Is there evidence that mice lacking Efnb1 in POMC neurons exhibit reduced glutamatergic input using electrophysiology?

The Discussion on POMC neurons in glucose-sensing is too speculative and not supported by the current studies.

Reviewer #3: In the manuscript by Dr Gervais and colleges the authors examine a mechanism for generating synaptic connections in the POMC and examine the impact of these connections on glucose homeostasis. Overall this is an interesting study that provides new information in an important area of research. There are a few issues that need to be addressed.

1. The paper lacks clear explanation for how some experiments are conducted. For instance in Figure 3, it is unclear whether the eGFP labeled cells are the ones where ephrin has been knocked down. 

2. In figure 3, to define synapses the authors need to stain for both pre (vGlut2) and postsynaptic markers (glutamate receptors or scaffolding proteins). 

3. It is unclear whether the authors are proposing ephrin-Bs are located in axons or on the postsynaptic cell. For glutamatergic synapse formation in the CNS, ephrin-Bs are typically axonal. It appears in figure 3 that the green neurites shown are axons. The authors should determine whether this is the case by staining for tuj1 or other axonal markers. This would help to clarify their model and story.

4. The authors propose that in the ephrin-B knockout mice that there are reduced number of excitatory synapses. However, the authors only show a decrease in vGlut2 puncta. Given the model that changes in both AMPAR and NMDAR levels may be needed, the authors should stain for these markers in the ephrin-B knockout mice. This would help to determine whether the authors model is correct. 

5. The sex dependent differences are interesting. The authors should address whether the pattern of expression of ephrin-Bs or the impact on vGlut2 puncta differs between male and female animals. 

Minor

The authors often make statements without references, particularly when describing the function of ephrins and Ephs. A careful review of the manuscript for missing references in needed.

---

## [Decision Letter · Decision Letter 2]

6 Oct 2020

Dear Dr Croizier,

Thank you for submitting your revised Research Article entitled "EphrinB1 modulates glutamatergic inputs into POMC-expressing progenitors and controls glucose homeostasis" for publication in PLOS Biology. I have now obtained advice from the original reviewers and have discussed their comments with the Academic Editor. Please accept my apologies for the delay in sending the decision below to you.

Based on the reviews, we will probably accept this manuscript for publication, assuming that you will modify the manuscript to address the remaining points raised by the reviewers. Having discussed reviewer 3's specific comments with the Academic Editor, we think you could add to the manuscript the limitation of the in vitro experimental model, as you did in the response to reviewers.

Please also make sure to address the data and other policy-related requests noted at the end of this email.

We expect to receive your revised manuscript within two weeks. Your revisions should address the specific points made by each reviewer. In addition to the remaining revisions and before we will be able to formally accept your manuscript and consider it "in press", we also need to ensure that your article conforms to our guidelines. A member of our team will be in touch shortly with a set of requests. As we can't proceed until these requirements are met, your swift response will help prevent delays to publication.

- a cover letter that should detail your responses to any editorial requests, if applicable

*Copyediting*

*Published Peer Review History*

*Early Version*

Sincerely,

Gabriel Gasque, Ph.D.,

Senior Editor,

ggasque@plos.org,

PLOS Biology

ETHICS STATEMENT:

-- Please include the ID number of your procedures approved by the Veterinary Office of Canton de Vaud

-- Please include the specific national or international regulations/guidelines to which your animal care and use protocol adhered. Please note that institutional or accreditation organization guidelines (such as AAALAC) do not meet this requirement.

DATA POLICY:

Note that we do not require all raw data. Rather, we ask for all individual quantitative observations that underlie the data summarized in the figures and results of your paper. For an example see here: http://www.plosbiology.org/article/info%3Adoi%2F10.1371%2Fjournal.pbio.1001908#s5

These data can be made available in one of the following forms:

Regardless of the method selected, please ensure that you provide the individual numerical values that underlie the summary data displayed in the following figure panels: Figures 1CF-K, 2AC, 3BD, 4CD-F, 5BC, 6CD, 7A-FHJ, 8ABD-M, S1E-G, S3A-D, S4B-K, and S5A-J.

Please also ensure that each figure legend in your manuscript includes information on where the underlying data can be found and that your supplemental data file/s has/have a legend.

Reviewer remarks:

Reviewer #1: The authors have responded satisfactorily to most comments. Yet, a few issues remain unresolved:

1) Line 145: The acknowledgement of P14-33 as an outlier suggests a distinction btw POMC->NPY and NPY->NPY, opposite to the authors' assertion.

2) Given the absolutely minimal overlap of Pomc-GFP +ve and Pomc-Cre-tdTom +ve cells), how do the authors know that the Pomc-GFP transgenic mouse recapitulates any native Pomc expression at P14? It is not clear what the authors mean by "during postnatal day 14, the number of arcuate Pomc neurons is more important to that observed in adults".

3) DAPI is conventionally blue. In Figure 1A (high magn.), nuclear staining is completely undiscernible. Images in Figures 4A, S1C are still missing DAPI. Figure 8C too. Nuclear staining in Fig. S4 is indicative of tissue deterioration. 

Reviewer #2: The authors have performed additional sutides which has addressed the Reviewers concerns in a satisfactory manner, which further strengthens the manuscript.

Reviewer #3: The authors have responded effectively to the majority of my comments and the comments of the reviewers. The manuscript is much improved. Unfortunately, the authors did not directly address my concerns about figure 3. I apologize if I was not clear - but my concerns remain. Here the authors show images of GFP labeled neurities and stain for vGlut2 a presynaptic marker. They show that there are decreases in this vGlut2 staining. These data would seem to suggest that in this figure, ephrin-B shRNAs are acting in the axon. However, for the rest of the manuscript, they claim that ephrin-Bs are acting postsynaptically. It seems that either the model system used in this figure is NOT appropriate for their studies (eg it looks at ephrin-B functions in AXONS, not dendrites) or ephrin-Bs are not functioning in axons in their studies. Based on the rest of their work, it seems likely that ephrin-Bs are postsynaptic in the POMC. This means that this assay is not a good model for their system. If the effects of ephrin-B are postsynaptic, the authors must look at postsynaptic markers to validate their shRNAs. In addition, no rescue controls are shown for these tools.

---

## [Editor Report · Decision Letter 3]

5 Nov 2020

Dear Dr Croizier,

On behalf of my colleagues and the Academic Editor, Rebecca Haeusler, I am pleased to inform you that we will be delighted to publish your Research Article in PLOS Biology. 

PRODUCTION PROCESS

Before publication you will see the copyedited word document (within 5 business days) and a PDF proof shortly after that. The copyeditor will be in touch shortly before sending you the copyedited Word document. We will make some revisions at copyediting stage to conform to our general style, and for clarification. When you receive this version you should check and revise it very carefully, including figures, tables, references, and supporting information, because corrections at the next stage (proofs) will be strictly limited to (1) errors in author names or affiliations, (2) errors of scientific fact that would cause misunderstandings to readers, and (3) printer's (introduced) errors. Please return the copyedited file within 2 business days in order to ensure timely delivery of the PDF proof. 

If you are likely to be away when either this document or the proof is sent, please ensure we have contact information of a second person, as we will need you to respond quickly at each point. Given the disruptions resulting from the ongoing COVID-19 pandemic, there may be delays in the production process. We apologise in advance for any inconvenience caused and will do our best to minimize impact as far as possible.

EARLY VERSION

PRESS 

Kind regards,

Erin O'Loughlin

Publishing Editor, 

PLOS Biology

on behalf of

Gabriel Gasque,

Senior Editor

PLOS Biology